# Activating reversible carbonate reactions in Nasicon solid electrolyte-based Na-air battery via in-situ formed catholyte

Heetaek Park[1], Minseok Kang[1], Donghun Lee[1], Jaehyun Park[2], Seok Ju Kang [2] & Byoungwoo Kang [1]✉

Out of practicality, ambient air rather than oxygen is preferred as a fuel in electrochemical systems, but $CO_2$ and $H_2O$ present in air cause severe irreversible reactions, such as the formation of carbonates and hydroxides, which typically degrades performance. Herein, we report on a Na-air battery enabled by a reversible carbonate reaction ($Na_2CO_3 \cdot xH_2O$, x = 0 or 1) in Nasicon solid electrolyte ($Na_3Zr_2Si_2PO_{12}$) that delivers a much higher discharge potential of 3.4 V than other metal-air batteries resulting in high energy density and achieves > 86 % energy efficiency at $0.1\,mA\,cm^{-2}$ over 100 cycles. This cell design takes advantage of moisture in ambient air to form an in-situ catholyte via the deliquescent property of NaOH. As a result, not only reversible electrochemical reaction of $Na_2CO_3 \cdot xH_2O$ is activated but also its kinetics is facilitated. Our results demonstrate the reversible use of free ambient air as a fuel, enabled by the reversible electrochemical reaction of carbonates with a solid electrolyte.

Extending from portable devices to electric vehicles and large-scale energy storage systems, the demand for Li-ion batteries with high energy density is continuously increasing and will overwhelm the pace of their development as the thermodynamic limits of Li-ion are reached. Even though metal-air batteries, utilizing lithium or sodium, have been of great interest owing to their exceptionally high theoretical gravimetric energy densities[1–4], these batteries predominantly rely on the utilization of pure oxygen for the formation and decomposition of metal oxides, rather than ambient air[5–7]. The use of pure oxygen in metal-air batteries causes several challenging problems because the chemical interactions occurring between the discharge products and atmospheric constituents such as $CO_2$ and $H_2O$ (Supplementary Table S1)[8] are inevitable leading to the formation of carbonates or hydroxides[9–11] that are notably challenging to decompose through electrochemical methods. Consequently, additional devices that can select and store a pure oxygen are required but these can substantially decrease promising gravimetric and volumetric energy densities. In metal-air batteries that are based on liquid or solid electrolytes, the

formation of carbonates is not desirable for reversible electrochemical performance and thus should be avoided if possible[12,13]. Since metal carbonates and hydroxides have high thermodynamic stability, they need much higher voltage for the decomposition than upper limit of electrochemical window in aprotic liquid electrolytes[13]. As a result, it causes poor round-trip energy efficiency (=$E_{discharge}/E_{charge}$), the ratio of input energy in the charge to output energy in the discharge[10,13]. In addition, the use of ambient air in metal–air batteries can cause severe side reactions with the cathodes that are composed of aprotic electrolytes and carbon, and can contaminate the anode[14] leading to poor electrochemical performance.

Several studies have sought to address the challenges of operating in ambient air by using an all-solid-state Li-air battery design with oxide-based solid electrolytes that are chemically stable against air and have a wide electrochemical potential window[13,15–18]. Even though lithium oxides and hydroxides are electrochemically formed as discharge products in all-solid-state Li-air cells, these discharge products can still react chemically with $CO_2/H_2O$ in air to form lithium

[1]Department of Materials Science and Engineering, Pohang University of Science and Technology (POSTECH), 77 Cheongamro, Namgu, Pohang, Gyeongbuk 37673, South Korea. [2]Department of Energy Engineering School of Energy and Chemical Engineering Ulsan National Institute of Science and Technology (UNIST) 50 UNIST-gil, Ulsan 44919, South Korea. ✉e-mail: bwkang@postech.ac.kr

hydroxides and carbonates. Since the oxide-based solid electrolyte has a wide electrochemical window, the hydroxides/carbonates can be electrochemically decomposed in the charge process by applying a high over-potential[13] leading to the increase in coulombic efficiency. However, it can cause poor energy efficiency because different electrochemical reactions occur during charge (hydroxide and/or carbonate reactions) and discharge (oxides and/or hydroxide reactions) (Supplementary Table S2), and thereby large potential gap between charge and discharge cannot be eliminated. Even though a quasi-solid-state Na-air cell, which has a Nasicon SE with anolyte and an ionic liquid gel electrolyte, was recently reported[19], there have not been any reports yet on the all-solid-state Na-$O_2$/air batteries in part because of difficulties in their fabrication, and high interfacial resistance in using oxide-based solid electrolytes caused by the lack of a triple phase boundary (TPB), which is an active reaction area between a gas (air), an electronic conductor, and an ionic conductor.

Here, we report on the use of ambient air as a fuel in a Nasicon ($Na_3Zr_2Si_2PO_{12}$) solid electrolyte(SE)-based Na-air battery requiring no additional devices by exploiting reversible $Na_2CO_3 \cdot xH_2O$ (x = 0 or 1) reactions during cycling, leading to the highest operating voltage (~3.4 V) among metal-air batteries based on metal oxides or carbonate/hydroxide reactions. In contrast to previous reports, moisture in air reacts with the discharge products such as NaOH leading to an in-situ formed catholyte, acting as both an electrolyte and an active material, and the in-situ formed catholyte unexpectedly activates reversible carbonate reactions in a large active reaction area, leading to increased achievable capacity and enhanced electrochemical kinetics. The formation of the catholyte as the product of reaction with ambient air allows the cell to undergo the same electrochemical reaction pathway with sodium carbonates in both the charge and discharge without forming any intermediate phases. As a result, the potential gap between charge and discharge reaction shrinks, leading to high round-trip energy efficiency. The resulting solid-state Na-air battery delivers high reversible energy density for 100 cycles with high coulombic and energy efficiencies.

## Results and discussion
### Fabrication of Nasicon solid electrolyte-based Na-air battery
Utilizing ambient air as a fuel source, we constructed Na-air cells based on Nasicon solid electrolyte (SE), incorporating an air-electrode that comprises Nasicon ($Na_3Zr_2Si_2PO_{12}$) SE as an ionic conductor and nickel (Ni) metal instead of carbon as the electronic conductor (Fig. 1a–d). This design can be realized by the superior chemical stability of Nasicon SE in ambient air[20]. Previous reports studying Na-air batteries with

hybrid electrolytes (aqueous and SE) clearly demonstrate that the side reactions between Nasicon and the other components ($H_2O$, $O_2$, and $CO_2$) are not observed, and reliable electrochemical performance was possible even though dense Nasicon SE was in direct contact with $H_2O$ as well as flowing gases such as $O_2$ and $CO_2$[21,22]. Moreover, the employment of solid electrolytes can help suppress the chemical reactions between the air and the Na metal anode and the use of the carbon-free air-electrode can mitigate unpredictable side reactions, commonly induced by carbon (Supplementary Table S1)[14]. Dense Nasicon SE was synthesized by a solid-state reaction and had ionic conductivity of ~0.2 mS cm$^{-1}$ at room temperature (RT) (Supplementary Fig. S1). To reduce the interfacial resistance between dense solid electrolyte and the air-electrode, and increase the activity in the air-electrode, a duplex solid electrolyte that is composed of the dense solid electrolyte and a porous solid electrolyte was prepared. The dense solid electrolyte was firstly prepared and then the porous Nasicon layer was fabricated on the dense Nasicon by a screen-printing process using a Nasicon slurry with a pore former (Di-ethylene glycol butyl ether). After this, the duplex (porous + dense) solid electrolyte (Fig. 1a, b) was heated at 1100 °C to sinter the Nasicon slurry and enhance contact between porous and dense Nasicon SE. For electron conduction, Ni nanoparticles were formed inside the porous Nasicon of the duplex structure by an infiltration process using $Ni(NO_3)_2 \cdot 6H_2O$ aqueous solution (Fig. 1b). Finally, a porous Ni current collector was laminated on the porous Nasicon with infiltrated Ni nanoparticles by the screen-printing process with Ni slurry containing the pore former (Fig. 1a). The Ni current collector was chosen because it has relatively high electrochemical stability with water compared to other metals; no reactions or changes in chemical nature of the air-cathode (Nasicon + Ni metal) were observed even after linear sweep test (Supplementary Fig. S2). The resulting assemblage (Fig. 1a, d) had ~50 μm thickness of the air-electrode, comprised of porous Nasicon, Ni nanoparticle, and porous Ni layer as a current collector.

To reduce interfacial resistance on the anode side, a thin gold (Au) layer (thickness ~10 nm) was deposited between Na metal and the dense Nasicon SE (Fig. 1a, e) using an ion coater. The interfacial resistance significantly decreased (Supplementary Fig. S3) because the formation of Na-Au alloy can make a homogeneous contact at the interface[23–26]. The cell was assembled (Supplementary Fig. S4), and then its air-electrode side was purged with $O_2$ gas (99.995% purity) for 3 h before electrochemical test to increase the cell's integrity (Supplementary Fig. S5). The purged $O_2$ can induce the formation of a protection layer that can shield the Na metal anode. We speculate that ambient air, especially moisture, can cross over to the Na anode side

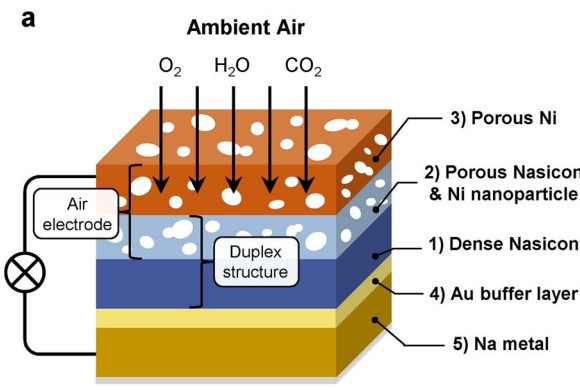
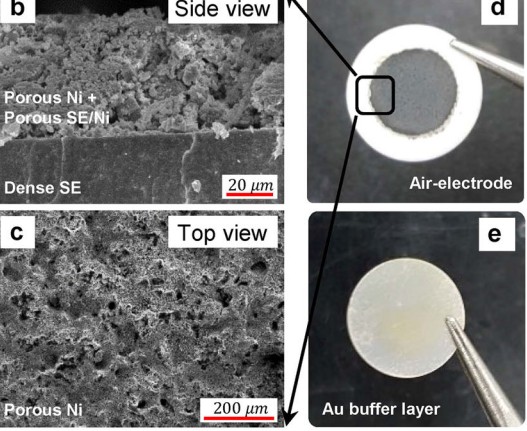

**Fig. 1 | Configuration of the Nasicon SE-based Na-air cell by using a duplex structure with a dense/porous Nasicon solid electrolyte. a** Schematic diagram of the Nasicon SE-based Na-air cell with carbon-free air-electrode and the duplex solid electrolyte. Microstructures of the SE-based Na-air cell: (**b**) cross-sectional and (**c**) top view of the air-electrode. Optical images of (**d**) air-electrode and (**e**) anode side of the cell.

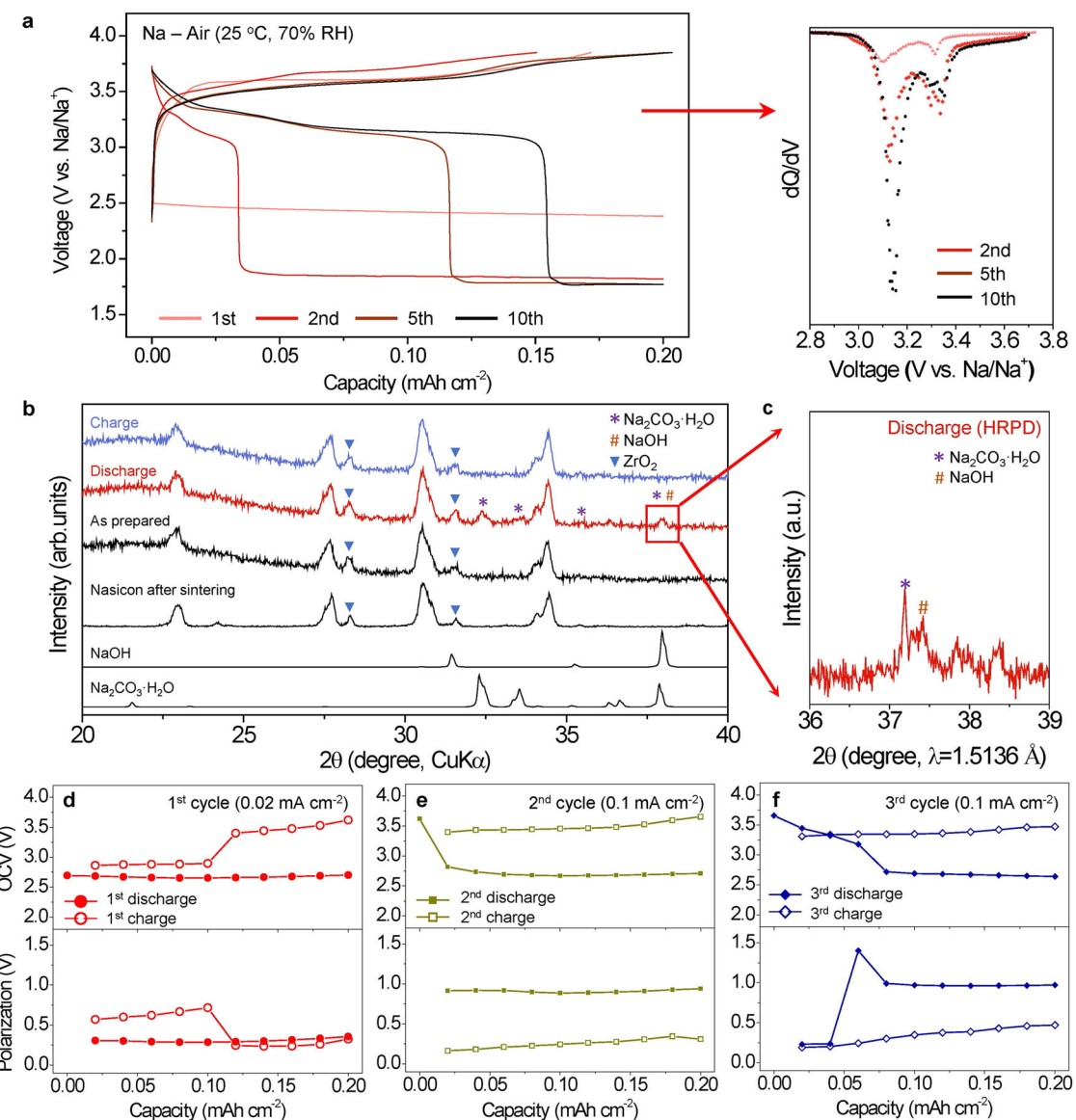

**Fig. 2 | Reaction products at 1st cycle and the electrochemical reactions via GITT measurements in the Nasicon SE-based Na-air cell. a** Voltage profiles (left) and differential discharge capacity plots, $dQ/dV$ for the 2nd to 10th cycles (right) during the first 10 cycles under the ambient air condition with cutoffs set at 0.2 mAh cm⁻² for discharge and 3.85 V for charge. The current densities were set at 0.02 mA cm⁻² for the first cycle and 0.1 mA cm⁻² for the subsequent cycles. **b** Ex-situ XRD patterns of the air-electrodes in a pristine state, after 1st discharge, and after 1st charge.

(Voltage profiles: Supplementary Fig. S7). **c** Ex-situ synchrotron X-ray powder diffraction of the discharged air-electrode after the 1st cycle. **d–f** OCVs of the cell (upper) and its polarization (lower) from the GITT measurements during (**d**) 1st cycle, (**e**) 2nd cycle, and (**f**) 3rd cycle. Current pulses of 0.02 mA cm⁻² (1st cycle) and 0.1 mA cm⁻² (2nd and 3rd cycles) for 12 min, followed by a rest for 1 h after each pulse were employed. All data in Fig. 2 were obtained by testing cells in open air with 70% RH at 25 °C.

## Understanding the electrochemical redox mechanism of SE-based Na-air battery

The Nasicon SE-based Na-air cell was electrochemically cycled for 10 cycles in open air with 70% relative humidity (RH) at 25 °C (Fig. 2a). The current densities were 0.02 mA cm⁻² at the 1st cycle, and increased to 0.1 mA cm⁻² in subsequent cycles. This indicates that ambient air can indeed be used as a fuel for reversibly operating the Na-air cell. It should be noted that the cell shows higher electrochemical activity in ambient air than in other gases without moisture ($O_2$, $CO_2$, $N_2$, and their mixtures) (Supplementary Fig. S6a) implying the importance of moisture in activating reversible electrochemical reaction with the ambient air. In the 1st cycle, the discharge proceeds through

electrochemical reaction with a voltage plateau at ~2.5 V, whereas the charge process shows a reaction with a voltage plateau at ~3.6 V. As cycle proceeds, additional voltage plateaus at ~3.2 and ~3.4 V in the discharge appear (dQ/dV curve in Fig. 2a) and then begin to increase in capacity. It was also confirmed that the redox potentials observed in cyclic voltammetry measurement are consistent with the voltage profile in the constant current condition (Supplementary Fig. S6b). After 10 cycles, the plateaus at 3.2 V and 3.4 V are almost saturated without further increasing the capacity. This indicates that the electrochemical reactions in subsequent cycles have changed from the 1st cycle in the Na-air cell with the ambient air.

To characterize the reaction products from the 1st cycle, ex-situ X-ray diffraction (XRD) measurements were performed on the two air-electrodes (Fig. 2b). One electrode was discharged to 0.8 mAh cm⁻² and the other was charged up to 3.85 V just after the discharge

(Supplementary Fig. S7). The XRD clearly shows that $Na_2CO_3 \cdot H_2O$ and NaOH in the discharged electrode are formed and then their content diminishes due to electrochemical decomposition during charge. Synchrotron high resolution X-ray powder diffraction (Fig. 2c) further confirms the co-existence of NaOH and $Na_2CO_3 \cdot H_2O$ in the discharged electrode by observing the separation of the peak at ~37° in XRD. Ex-situ Raman spectra of the air-electrodes after discharge and charge in 1st cycle further confirms that the formation and decomposition of $Na_2CO_3 \cdot H_2O$ (x = 0 or 1) in the SE-based Na-air cell (Supplementary Fig. S8). In-situ Differential electrochemical mass spectrometry (DEMS) analysis was also performed to confirm decomposition of $Na_2CO_3 \cdot H_2O$ (Supplementary Fig. S9). In-situ DEMS clearly shows that a $CO_2$ gas is evolved during charge. Considering gas evolution rate based on the reaction, some of the evolved $CO_2$ is stored in the air-electrode (especially in the absorbed $H_2O$).

To take a closer look at the electrochemical reactions upon cycling in the Nasicon SE-based Na-air cell, galvanostatic intermittent titration technique (GITT) measurements were performed at 25 °C under open air of 70% RH for three cycles (Fig. 2d–f). The applied current density was 0.02 mA cm$^{-2}$ during the 1st cycle and increased to 0.1 mA cm$^{-2}$ during the 2nd and 3rd cycles. All currents were applied for 10 pulses with durations of 1 h during the 1st cycle and with durations of 12 min during the 2nd and 3rd cycles, followed by a rest for 1 h after each pulse. Open circuit voltages (OCVs) of the Na-air cell in the 1st cycle (Fig. 2d) were quite different from those in the subsequent cycles. This clearly indicates that different electrochemical redox reactions occur in the 1st cycle compared to the subsequent cycles, and the discharge products can vary during cycling. In the 1st discharge, a single discharge voltage plateau appears at ~2.7 V, whereas there are two main voltage plateaus during the 1st charge: one at ~2.7 V, which can be the corresponding reaction of the observed discharge redox reaction, and another at ~3.4 V, which may be an additional redox reaction. These two distinct voltages in the 1st charge suggest that two Na compounds can form as reaction products during or after the 1st discharge. Given that both $Na_2CO_3 \cdot H_2O$ and NaOH in the discharged electrode are observed in XRD (Fig. 2b, c) and Raman spectra (Supplementary Fig. S8), and only a single electrochemical redox reaction at ~ 2.7 V in the discharge is observed in the 1st discharge, only one of the two Na compounds is formed electrochemically during the 1st discharge while the other is not. Considering the thermodynamic redox potential of NaOH (Supplementary Table S2) and the relatively high concentration of $O_2$ and $H_2O$ in ambient air, NaOH could be electrochemically formed at ~2.7 V during the 1st discharge (Eqs. (1) and (2)). The following reaction describes the origin of the single voltage plateau observed.

At the Na electrode side:

$$Na \leftrightarrow Na^+ + e^- \tag{1}$$

At the air-electrode side:

$$Na^+ + e^- + 0.5H_2O(g) + 0.25O_2(g) \leftrightarrow NaOH(s), E = 2.75 V \, vs. \, Na/Na^+ \tag{2}$$

Noting that two Na compounds were observed in the 1st discharge and sodium carbonates have higher thermodynamic stability than NaOH (Supplementary Table S2), the formation of $Na_2CO_3 \cdot H_2O$ may be caused by chemical reactions of NaOH with $CO_2$ and $H_2O$ from the ambient air during or after the 1st discharge by following the below reactions (Eqs. (3) and (4)):

$$2NaOH + CO_2 \rightarrow Na_2CO_3 + H_2O \tag{3}$$

$$2NaOH + CO_2 \rightarrow Na_2CO_3 \cdot H_2O \tag{4}$$

Therefore, the two plateaus observed in the 1st charge (Fig. 2d) can be ascribed to the electrochemical decomposition of NaOH at ~2.7 V (Eqs. (1) and (2) and $Na_2CO_3 \cdot xH_2O$ (x = 0 or 1) at ~3.4 V (Eqs. (1), (5) and (6)), respectively.

At the air-electrode side:

$$2Na^+ + 2e^- + CO_2(g) + 0.5O_2(g) \leftrightarrow Na_2CO_3(s), E = 3.37 V \, vs. \, Na/Na^+ \tag{5}$$

$$2Na^+ + 2e^- + CO_2(g) + 0.5O_2(g) + H_2O(g) \\ \leftrightarrow Na_2CO_3 \cdot H_2O(s), E = 3.43 V \, vs. \, Na/Na^+ \tag{6}$$

The OCVs of the first charge plateau slightly increases from ~2.7 V to ~2.9 V (Fig. 2d). Considering that NaOH·$H_2O$ tends to be formed when NaOH is exposed in humid air[27], we speculate that NaOH·$H_2O$ is also chemically formed at the end of discharge and it can lead to the increase in the OCVs of the first charge plateau (see NaOH·$H_2O$ reactions in Supplementary Table S2). After the 1st cycle, the additional redox reaction at ~3.4 V in subsequent discharge cycles starts to grow and becomes dominant, whereas the redox reaction at ~2.7 V in subsequent charge cycles disappears after the 1st cycle (Fig. 2e, f). Given the thermodynamic potentials and the differential capacity (d$Q$/d$V$) plot (Fig. 2a), the voltage plateau at ~3.4 V can be a result of the formation of sodium carbonates such as $Na_2CO_3 \cdot xH_2O$ (x = 0 or 1) (Eqs. (5) and (6)).

To understand the discharged products in the SE-based Na-air cell, ex-situ measurements in the air-electrodes that were pre-cycled for 10 cycles were performed. (Supplementary Fig. S10). Electrochemical formations of $Na_2CO_3 \cdot xH_2O$ at ~3.4 V and NaOH at ~2.0 V were confirmed by using ex-situ XRD and Raman spectroscopy measurements. When the SE-based Na-air cell was discharged to 2.8 V (Supplementary Fig. S10a), the formation of the discharge products ($Na_2CO_3 \cdot xH_2O$) was clearly observed by ex-situ Raman spectroscopy. Raman measurement clearly shows that the $CO_3^{2-}$ band of $Na_2CO_3$ and OH bands of $H_2O$ was detected in the cell discharged to 2.8 V. In-situ Raman analysis also shows the direct formation of $Na_2CO_3 \cdot xH_2O$ during discharge (Supplementary Fig. S11). When the discharge process was further proceeded to 0.8 mAh cm$^{-2}$ and to 1.5 mAh cm$^{-2}$ at ~2.0 V, the intensity of XRD peaks corresponding to NaOH in the ex-situ air-electrodes continuously increased compared to those of $Na_2CO_3 \cdot xH_2O$ and Nasicon. This indicates that the reaction at ~2.0 V is ascribed to the electrochemical formation of NaOH. Combining these results, we can conclude that the $Na_2CO_3 \cdot xH_2O$ and NaOH is electrochemically formed at high (~3.4 V) and low (~2.0 V) potential, respectively. The electrochemical reaction of $Na_2CO_3 \cdot xH_2O$ in the Na-air cell is activated and becomes the predominant redox reaction as cycling proceeds. This is the first report of a reversible reaction via the electrochemical formation of carbonate compounds in the Li/Na-ambient air cells. It should be noted that the electrochemical decomposition/formation of $Na_2CO_3 \cdot xH_2O$ (Eqs. (5) and (6)) in the Na-air cell occurs at much higher redox potential than reported carbonate reactions[11,28–30] partly because the sodium carbonate reactions in the Na-air cell can occur directly without any intermediate phases or reactions.

Surprisingly, the electrochemical reactions of the $Na_2CO_3 \cdot xH_2O$ exhibit much smaller polarization than that of the NaOH. In the 1st cycle, tested at 0.02 mA cm$^{-2}$, the polarizations of the NaOH redox reaction were ~0.3 V for the discharge and ~0.6 V for the charge, whereas the polarization of the $Na_2CO_3 \cdot xH_2O$ reactions occurred at ~3.4 V was ~0.25 V for the 1st charge (Fig. 2d). Also, even at increased the current density of 0.1 mA cm$^{-2}$ after 1st cycle (five times higher than the 1st cycle), the polarization of the $Na_2CO_3 \cdot xH_2O$ decomposition/formation reactions occurred at ~3.4 V in the charge/charge in Fig. 2e, f was almost similar to that of the 1st cycle. This indicates that the electrochemical decomposition/formation reactions of $Na_2CO_3 \cdot xH_2O$ can be

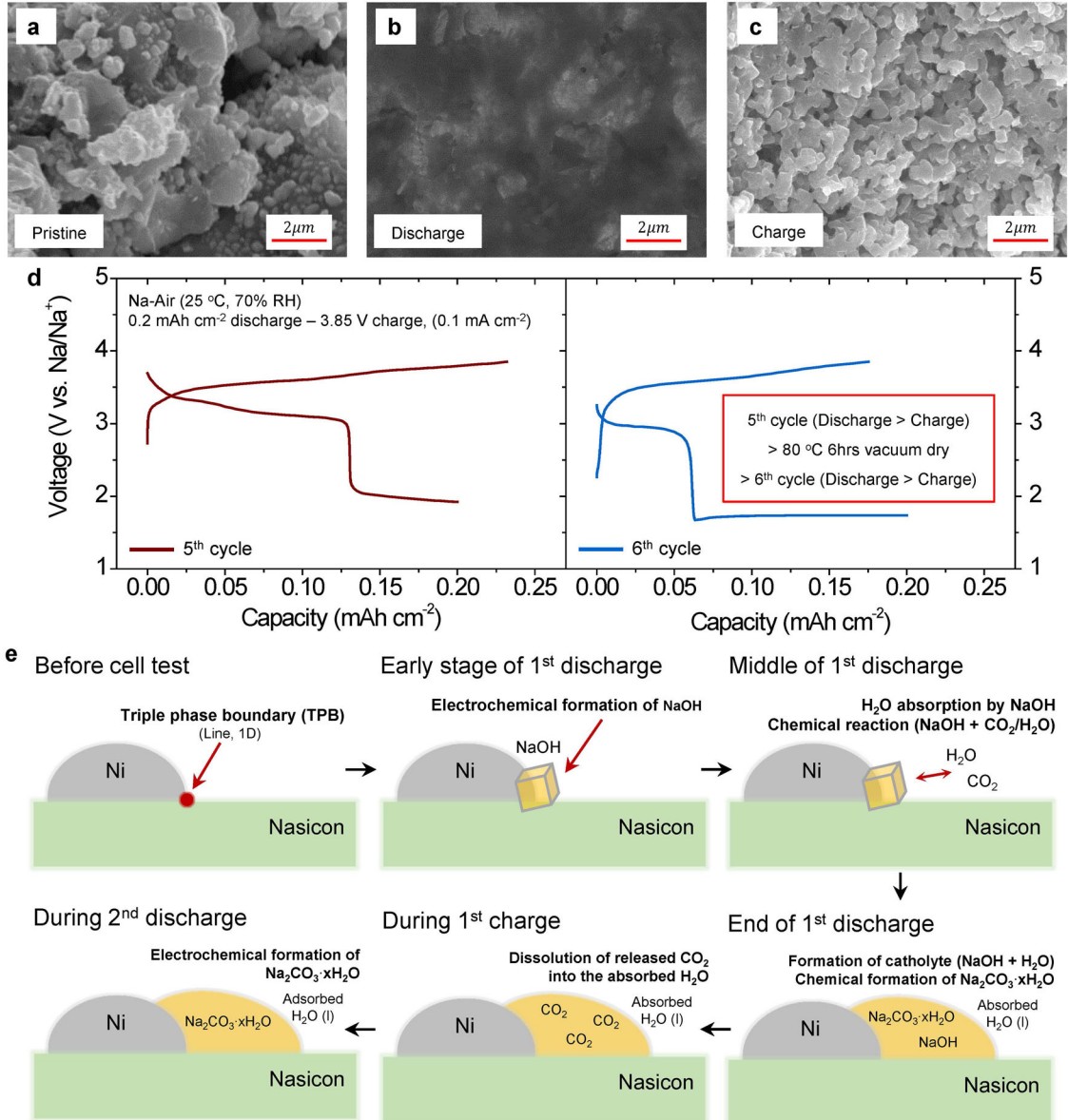

**Fig. 3 | Effects of the absorbed H₂O on electrochemical reactions in the Nasicon SE-based Na-air cell.** SEM images (cross-section view) of the air-electrodes (**a**) in the pristine, (**b**) after 1$^{st}$ discharge (0.8 mAh cm$^{-2}$), and (**c**) after the 1$^{st}$ charge (0.8 mAh cm$^{-2}$ discharge and 3.85 V charge). **d** Voltage profiles of 5$^{th}$ cycle, and 6$^{th}$ cycle (0.2 mAh cm$^{-2}$ discharge and 3.85 V charge cutoff conditions) followed by a vacuum drying at 80 °C for 6 h after 5$^{th}$ charge. The current density was 0.1 mA cm$^{-2}$. **e** Schematic diagram of the activation of electrochemical Na₂CO₃ · xH₂O reactions in the Nasicon SE-based Na-air cell with the ambient air. Data in Fig. 3 were obtained by testing the cells in open air with 70% RH at 25 °C.

kinetically facile, unlike previous results[9,13]. In contrast, the NaOH reaction shows sharp increase in the polarization from approximately 0.3 V at 0.02 mA cm$^{-2}$ to about 0.9 V at 0.1 mA cm$^{-2}$ (Fig. 2d, e, f). Sodium carbonate's electrochemical reactions in Nasicon SE-based Na-air cells are substantially activated and improved through a thermodynamic reaction pathway during the charging/discharging process. As a result, the Na-air battery with the ambient air in subsequent cycles will deliver higher redox potential (-3.4 V) than Na-O₂ cell[5,6], and achieve low polarization even with main electrochemical reactions of sodium carbonates.

### Effects of the absorbed H₂O on electrochemical reactions in the SE-based Na-air cell

To further characterize electrochemical/chemical reactions during cycling, we carried out scanning electron microscopy (SEM) measurements for the air-electrodes of two cells. One was discharged to 0.8 mAh cm$^{-2}$ and the other was charged to 3.85 V after the discharge (Supplementary Fig. S7). The air-electrodes of these two cells were compared with the pristine cell. Before the cell test, the air-electrode was observed as a particulate form (Fig. 3a and Supplementary Fig. S12a). However, the discharge products in the air-electrode showed a film-like morphology that covers the entire electrode after 1$^{st}$ discharge (Fig. 3b and Supplementary Fig. S12b) indicating that the electrochemical/chemical reaction is widespread in the electrode. Most of the film-like discharge products disappeared after 1$^{st}$ charge. It can demonstrate that the discharge products can be electrochemically decomposed (Fig. 3c and Supplementary Fig. S12c). However, the morphology of particles in the charged cell is still slightly blurred compared to that of the pristine cell. It might be originated from a residual water because the charge reaction is not complete during 1$^{st}$ charge (Supplementary Fig. S7). We also note that the discharge products electrochemically formed and decomposed even in 10$^{th}$ cycle

(Supplementary Fig. S13) revealing the reversibility of the electrochemical reactions via the formation of sodium carbonates with NaOH (Eqs. (2), (5) and (6)). Furthermore, the film-like morphology clearly shows that the reacted products in the 1st discharge can be formed through the absorption of liquid reactants such as $H_2O$ from the ambient air. Given that NaOH, electrochemically formed during 1st discharge, has a strong deliquescent property, it easily absorbs $H_2O$ from the ambient air until the NaOH dissolves into the absorbed $H_2O$[9] leading to the formation of catholyte. The film-like morphology of the reacted products can be the result of the in-situ formed catholyte. The existence of $H_2O$ in the air-electrode was further confirmed by electrochemical test (Supplementary Fig. S14a, b). Upon discharging the cell to 0.2 mAh cm$^{-2}$ and subsequently charging it to exceed 3.9 V, a considerable increase in capacity was observed around 3.9 V, which is similar to the thermodynamic potential of a water decomposition[21,22]. This increased capacity suggests that $H_2O$ can stay within the air-electrode even after the initial charge unless a higher voltage (exceeding approximately 3.9 V) is applied. Such residual water can easily form a blurred boundary between particles even after charging to <3.9 V. Removal of the film-like morphology by performing a vacuum drying of the discharged cell further supports that the blurred particle boundaries are formed by residual water (Supplementary Fig. S14c). Therefore, the formation of a catholyte through chemical reaction with the discharged products of the SE-based Na-air battery is facilitated by the moisture in ambient air. This reuslt is quite different from previous Na-air reports, which show a deteriorated electrochemical reaction under the existence of moisture[9].

The in-situ formed catholyte simultaneously facilitate both the chemical formation of $Na_2CO_3 \cdot xH_2O$ and the electrochemical activation of the $Na_2CO_3 \cdot xH_2O$ reactions. Considering that a $CO_2$ concentration in the ambient air is not possible to activate the electrochemical discharge reactions of $Na_2CO_3 \cdot xH_2O$, it would be reasonable for thinking that $CO_2$ gas, generated through the electrochemical decomposition of the $Na_2CO_3 \cdot xH_2O$ (Eqs. (5) and (6)) during charge, might not be released into the ambient air but rather retained within the air-electrode, particularly close to the reaction sites. The locally increased retention of the $CO_2$ gas within the air-electrode can be caused by the absorbed $H_2O$ because $CO_2$ is significantly more soluble in $H_2O$ than $O_2$[31]. This is further supported by in situ Differential Electrochemical Mass Spectrometry (DEMS) analysis, which clearly demonstrates that the actual amount of evolved $CO_2$ during charge is less than the theoretical amount calculated by Faraday's law of electrolysis (Supplementary Fig. S9). This indirectly shows the $CO_2$ dissolution by the absorbed $H_2O$ in the air-electrode. As a result, the increase in the local concentration of $CO_2$ gas in the air-electrode promotes the electrochemical formation of $Na_2CO_3 \cdot xH_2O$ (Eqs. (5) and (6)) in the subsequent discharge cycles. It should be noted that side reactions related with Ni nanoparticles ($Ni(OH)_2$/NiOOH) might be exhibited at around this potential[32,33] but this effect was negligible due to lack of the electrochemical activity (Supplementary Fig. S15). This indicates that the electrochemical reactions in solid electrolyte Na-air cell can be dominated by sodium carbonates and sodium hydroxides. Considering that the electrochemical formation and decomposition reactions of $Na_2CO_3 \cdot xH_2O$ continuously increase for 10 cycles via in-situ formed catholyte and then is saturated, the electrochemical decomposition of $Na_2CO_3 \cdot xH_2O$ can keep increasing the amount of $CO_2$ gas inside the air-electrode but finally is saturated.

To understand the effect of the absorbed $H_2O$ in the air-electrode on the activation of the $Na_2CO_3 \cdot xH_2O$ electrochemical reactions, the Na-air cell was dried up by a vacuum drying at 80 °C for 6 h after the charge (Fig. 3d) to remove $H_2O$ inside the air-electrode. Before performing the vacuum drying process, the cell was cycled for five times to sufficiently activate the electrochemical reactions of $Na_2CO_3 \cdot xH_2O$. The current densities were 0.02 mA cm$^{-2}$ at the 1st cycle and 0.1 mA cm$^{-2}$ at the other cycles. After the activation process, the cell at the end of 5th charge was

dried up at 80 °C for 6 h under vacuum condition. This vacuum drying process severely reduced the discharge capacity of the $Na_2CO_3 \cdot xH_2O$ from 0.13 mAh cm$^{-2}$ at 5th cycle to 0.06 mAh cm$^{-2}$ at 6th cycle. The result indicates that the loss of the absorbed $H_2O$ in the air-electrode strongly reduces the electrochemical formation of $Na_2CO_3 \cdot xH_2O$ partly due to a decrease in the amount of the in-situ formed catholyte, which can lower of the $CO_2$ concentration nearby the reaction sites inside the air-electrode. This demonstrates that the existence of the absorbed $H_2O$ significantly affects for the activation of the electrochemical reaction of $Na_2CO_3 \cdot xH_2O$.

To further understand the origin of the carbonate reactions, the cells were constructed with the 'hybrid (aqueous electrolyte with NaOH or $Na_2CO_3$ and Nasicon solid electrolyte)' electrolyte system; it is similar to the discharged state of the Nasicon SE based Na-air battery with the in-situ formed catholyte. The cell with the hybrid electrolyte was prepared by filling the aqueous solution (1 mL) to the air-cathode side. In addition, the air-electrode side of the cell was closed to suppress water evaporation during cell tests (Supplementary Fig. S16a) unlike the Na-air cell, which is open to the ambient air. Electrochemical tests were carried out at RT. The ~3.4 V discharge plateau was barely increased by cycling the cell with the NaOH (aq) hybrid electrolyte (Supplementary Fig. S16b). On the contrary, the cell with the $Na_2CO_3$ (aq) hybrid electrolyte showed the significant increase of the ~3.4 V plateau and exhibited a similar voltage curve to the Nasicon SE based Na-air cell operating in the ambient air (Supplementary Fig. S16c). To further understand the role of $CO_2$ for the electrochemical formation/decomposition of $Na_2CO_3 \cdot xH_2O$, the SE-based $Na-O_2$ bubbled $H_2O$ cell, which is $CO_2$-free cell, was prepared and then tested (Supplementary Fig. S16d). It did not show the ~3.4 V voltage reaction, and did not have the continuous activation of ~3.4 V reaction during cycles compared to the SE-based Na-air cell. In addition, Raman measurements were carried out in order to observe the discharge product in the cell with $Na_2CO_3$ (aq) hybrid electrolyte. Raman spectroscopy measurements of the electrodes (Supplementary Fig. S17) show that carbonates and hydroxides were formed at ~3.4 V and ~2.0 V, respectively during discharging the cell. Given that the electrochemical reactions in hybrid Na-air cell with $Na_2CO_3$ (aq) solution is quite similar with those of the SE-based Na-air cell, it indirectly supports that $Na_2CO_3$ is electrochemically formed at ~3.4 V in the SE-based Na-air cell. These results clearly demonstrate that the ~3.4 V reaction is related to the formation of $Na_2CO_3 \cdot xH_2O$ and the $CO_2$ from the air is an essential component for activating this carbonate reaction. Considering that electrochemical decomposition of $Na_2CO_3$ in the presence of water during charge can activate the electrochemical reaction at ~3.4 V in the subsequent discharge, it can be demonstrated that the carbonate reactions in the Nasicon SE-based Na-air cell can be activated by the dissolution of $CO_2$ in $H_2O$. In consequence, the in-situ formed catholyte in the Nasicon SE-based Na-air cell during cycles critically affects the activation of the $Na_2CO_3 \cdot xH_2O$ reactions and its reversible electrochemical reaction.

Figure 3e shows a schematic diagram describing the activation of electrochemical reactions with $Na_2CO_3 \cdot xH_2O$ and their reversibility in the SE based Na-air cell with the ambient air. At the beginning of the 1st discharge, NaOH is formed electrochemically through reacting with $O_2/H_2O$ from the ambient air at the triple phase boundary of Nasicon, Ni metal, and air (Eq. (2)). The discharge product, NaOH, spontaneously absorbs $H_2O$ from the ambient air until it is dissolved in the absorbed $H_2O$ leading to the formation of the catholyte. Then, the in-situ formed catholyte (NaOH +$H_2O$) can chemically react with $CO_2$ in the air to yield $Na_2CO_3$ $xH_2O$ during and after 1st discharge (Eqs. (3) and (4)). These discharged products can easily cover the entire electrode leading to the formation of the film-like morphology that can significantly increase the active reaction area due to in-situ formed catholyte. During 1st charge, the $Na_2CO_3$ $xH_2O$ and NaOH are electrochemically decomposed and then release $CO_2$ and $O_2$ or $H_2O$ (Eqs. (2), (5) and (6)). If the $CO_2$ and $O_2$ gases are evolved during 1st charge, the

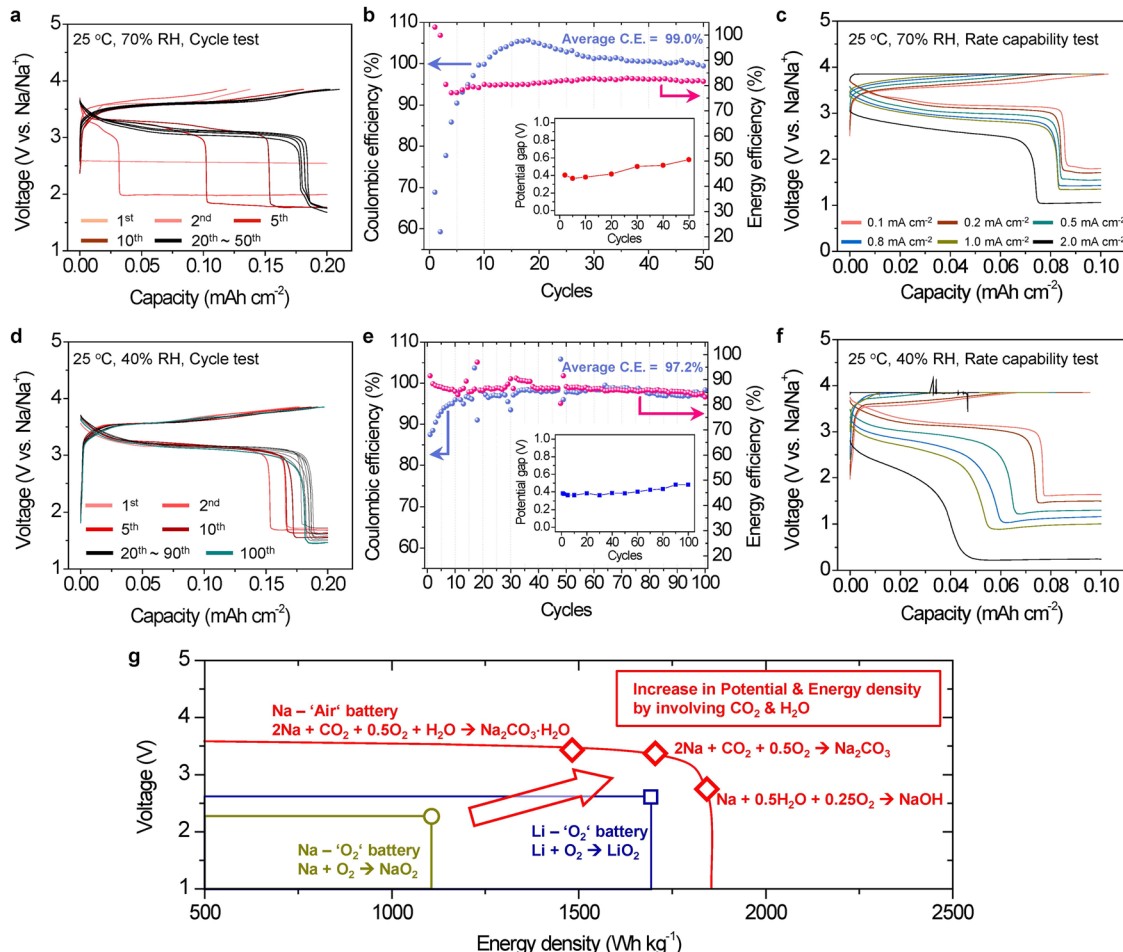

**Fig. 4 | Electrochemical performances of the Nasicon SE-based Na-air cells in ambient air at different RH levels. a–c** Cycle and Rate capability tests in open air with 70% RH at 25 °C: (**a**) Voltage profile and (**b**) efficiencies of the cycle test (inset: potential gap of the carbonates reaction between charge and discharge calculated from the d$Q$/d$V$ plots) at 0.1 mA cm$^{-2}$ within discharge and charge cutoffs of 0.2 mAh cm$^{-2}$ and 3.85 V, respectively. **c** Rate capability test conducted with discharge and charge cutoffs set at 0.2 mAh cm$^{-2}$ and 3.85 V, respectively, employing a 3.85 V CCCV mode until the current diminished to less than 30% of its initial value. **d–f** Cycle and Rate capability tests in open air with 40% RH at 25 °C, under the same current density and cutoff conditions as detailed in **a–c**: **d** Voltage profile and **e** efficiencies of the cycle test (inset: potential gap of the carbonates reaction between charge and discharge calculated from the d$Q$/d$V$ plots). Prior to these tests, the cell underwent a pre-activation process for 10 cycles at 70% RH and 25 °C (Fig. 2a). **f** Rate capability test following the activation process (Supplementary Fig. S21a). The legends in **f** are identical to those in **c**. **g** Potential-Energy density plot comparing the metal-O$_2$ cells based on superoxides with the SE-based Na-air cells.

absorbed H$_2$O might capture these gases (especially CO$_2$) nearby the reaction sites due to high CO$_2$ gas solubility in H$_2$O[31]. As a result, at the end of charge, CO$_2$ concentration at the inside of the air-electrode can increase drastically, and then in the following (2$^{nd}$) discharge the electrochemical formation of Na$_2$CO$_3$ xH$_2$O (Eqs. (5) and (6) can be activated. The appearance of multiple peaks in the d$Q$/d$V$ plot at ~3.4 V (Fig. 2a) is likely related not only to the electrochemical formation of Na$_2$CO$_3$ but also the change in OCV from the Na$_2$CO$_3$ xH$_2$O reactions (Eqs. (2), (5) and (6)) caused by various states (i.e. solid, gas, or aqueous) of reactants/products (Supplementary Table S2). During the 2$^{nd}$ discharge, the redox reactions consume most of the dissolved CO$_2$ gas in the catholyte and then subsequently, the electrochemical reaction of NaOH (Eq. (2)) occurs at the end of discharge. The formation of NaOH at the end of each discharge, which corresponds to the voltage plateau below 2 V after 2$^{nd}$ cycle, can lead to additional absorption of H$_2$O and CO$_2$ from the air at each cycle (Supplementary Fig. S18), which can enable to form the catholyte and help electrochemical decomposition of Na$_2$CO$_3$ xH$_2$O in subsequent charge processes. Also, kinetics of the chemical reactions between NaOH and CO$_2$ (Eqs. (3) and (4)) are also improved by the presence of the absorbed H$_2$O[9]. This allows most of the NaOH to form Na$_2$CO$_3$ xH$_2$O at the end of the 2$^{nd}$ and

3$^{rd}$ discharge as observed in the GITT test (Fig. 2e, f). As a result, the absolute amount of CO$_2$ gas inside the air-electrode can be increased, resulting in the gradual increase of the Na$_2$CO$_3$ xH$_2$O reactions in subsequent discharge cycles (Fig. 2a).

Furthermore, the in-situ formed catholyte can substantially improve other critical electrochemical properties of the Nasicon SE-based Na-air battery. The catholyte, which increases the active area, enables an increase of the achievable discharge capacity (Supplementary Fig. S19) by around 1.5 times (~6.3 mAh cm$^{-2}$) when compared to the maximum capacity calculated based on the pore volume in the air-electrode (Supplementary Table S3). In addition, an excess amount of the discharge product was found even on the Pt mesh (4 in Supplementary Fig. S4) that was on the porous Ni current collector (3 in Fig. 1a) and was not in contact with any ionic conductor (Supplementary Fig. S20). This result indicates that the in-situ formed catholyte can act as a new ionic conductor for increasing the active reaction area.

## Electrochemical performances of the SE-based Na-air cells in ambient air

The SE-based Na-air cell shows excellent electrochemical performances including superior capacity retention and high rate capability

under open air conditions with 70% RH (Fig. 4a–c) and 40% RH (Fig. 4d–f) due to the in-situ formed catholyte despite operating through the electrochemical reaction of carbonates and hydroxides.

In the cycle tests, the cells were discharged to 0.2 mAh cm$^{-2}$ and charged to 3.85 V. The current densities were 0.02 mA cm$^{-2}$ at 1$^{st}$ cycle and 0.1 mA cm$^{-2}$ in subsequent cycles. When the cell was tested in ambient air with 70% RH, the voltage plateau at ~3.4 V (corresponding to the $Na_2CO_3 \cdot xH_2O$ reactions) extended with increasing cycle number. This extension was saturated at the 10$^{th}$ cycle with the capacity of the plateau reaching ~0.16 mAh cm$^{-2}$ and the energy efficiency (=$E_{discharge}/E_{charge}$) converging to ~80% (Fig. 4a, b). It should be noted that the NaOH reaction at the end of discharge cycles still appears indicating the formation of the catholyte via the delinquency of NaOH can be sustained (Supplementary Fig. S18). After the activation of the carbonate reactions in the first 10 cycles, the cell tested in 70% RH showed excellent cycle stability. The cell showed 99.0% average coulombic efficiency and 82.1% average energy efficiency for 50 cycles. The coulombic efficiency between 10$^{th}$ and 25$^{th}$ cycle is higher than 100% (Fig. 4b), which may be attributed to residual discharge products forming between the 1$^{st}$ and 10$^{th}$ cycles where the efficiency is lower than 100%. After the activation process for 10 cycles, the kinetics of the Na-air cell are improved, allowing the residual discharge products to decompose during 10$^{th}$-25$^{th}$ cycles and thus the coulombic efficiency can be increased. The energy efficiency is much higher than the other metal-air cells in previous studies[3,6,8,11,13] because the electrochemical reaction pathways during charge and discharge are the same ($Na_2CO_3 \cdot xH_2O$ reactions) in the Na-air cell. High energy efficiency can be achieved since the potential gap of the $Na_2CO_3 \cdot xH_2O$ reactions between charge and discharge, ~0.4 V, is small (Fig. 4b). This illustrates that the Na-air cell operated in ambient air can be reversibly cycled via reversible $Na_2CO_3 \cdot xH_2O$ reactions and can increase energy efficiency.

The rate capability test was performed by discharging the cells with the current densities from 0.1 mA cm$^{-2}$ to 2.0 mA cm$^{-2}$ and charging the cells using a constant current - constant voltage (CCCV) protocol: charging to 3.85 V at constant current, and then applying a voltage hold at 3.85 V until the current reaches <30% of the applied current (Fig. 4c). The CCCV method was conducted to fully charge the cell without the decomposition of water above ~3.9 V (Supplementary Fig. S14).

Before the rate capability test, the cell was pre-activated for 10 cycles in ambient air with 70% RH (Supplementary Fig. S21a) to ensure that the $Na_2CO_3 \cdot xH_2O$ reaction was fully saturated. During the rate capability test, the cell was cycled five times at each current density (Supplementary Fig. S21b); only the voltage profile of the last cycle at each current density is shown (Fig. 4c). The cell could be operated at a high current density of 2.0 mA cm$^{-2}$ even though an increased polarization was observed. At 2.0 mA cm$^{-2}$ the coulombic efficiency was ~88%, and the energy efficiency was ~66%. This result clearly demonstrates that the electrochemical reactions involving carbonates and hydroxides in the Nasicon SE based Na-air cell are kinetically facile, partly due to the in-situ formed catholyte through chemical reaction of the discharged products with the air.

The cycle and rate capability tests were also performed in air with reduced RH, from 70% to 40% (Fig. 4d–f), in order to understand the effect of RH on the electrochemical performance. When the cell was cycled in air with only 40% RH, the $Na_2CO_3 \cdot xH_2O$ reactions were barely activated and increased even with repeated cycles (Supplementary Fig. S22a), and thereby the polarization was much higher than the cell in air with 70% RH. This indicates that the amount of $H_2O$ strongly affects the electrochemical activation of the carbonate reactions at the beginning cycles. To activate the electrochemical reactions of $Na_2CO_3 \cdot xH_2O$ in ambient air with 40% RH, the SE-based Na-air cell was pre-cycled in air with 70% RH for 10 cycles (Fig. 2a and Supplementary Fig. S21a). After pre-cycling the cell in air with 70% RH, the cycle and rate capability tests in air with 40% RH were carried out. The electrochemical properties of the cells in air with 40% RH (Fig. 4d) were comparable to those in air with 70% RH (Fig. 4a). This result implies that once the sufficient amount of the catholyte is formed and the sodium carbonate reaction is activated in the air-electrode under the air with high RH, the reversible $Na_2CO_3 \cdot xH_2O$ reaction is well maintained even in air with low RH in subsequent cycles. Surprisingly, the cell with 40% RH showed significantly improved cycle stability compared to the cell with 70% RH (Fig. 4e); the cell with 40% RH air showed stable capacity retention to 100 cycles with 97.2% average coulombic efficiency and 86.5 % average energy efficiency. Also, the potential gap between the charge and discharge during the cycle test (~0.4 V) was similar to that of the cell in air with 70% RH (Fig. 4e). Long extended cycle stability indicates that the reversible electrochemical reactions of $Na_2CO_3 \cdot xH_2O$ can be sustained even with less amount of $H_2O$ in air if the cell is fully activated by pre-cycling in air with high RH.

Furthermore, the cell in air with 40% RH also showed reasonable rate capability (Fig. 4f and Supplementary Fig. S22b). The cell was cycled up to a high current density of 2.0 mA cm$^{-2}$ even though it caused higher polarization than the cell with 70% RH. Also, the portion of the $Na_2CO_3 \cdot xH_2O$ reactions during the discharge decreased more rapidly as the current density increased, compared to the cell in air with 70% RH. We further confirmed that the SE-based Na-air cells after 10 cycles for the activation can be operated under ambient air with low RH from 20% to 40%. It was obviously observed that the cell performances (energy/coulombic efficiencies) decreased by lowering RH (Supplementary Fig. S23).

The cells with 70% RH showed the degradation in the capacity retention after ~50$^{th}$ cycles (Fig. 4b). When the Na metal was replaced by new metal, the cell was recovered and showed a typical voltage curve of the $Na_2CO_3 \cdot xH_2O$ reaction (Supplementary Fig. S24). This result strongly suggests that the degradation can be mainly originated from the corrosion of the Na metal rather than the air-electrode. Since the cycle life of the cells was significantly extended by operating them in air with low RH = 40% (Fig. 4e), so we speculate that influx of air (especially $H_2O$ in it) in cycles into the Na metal anode might severely degrade the Na metal.

The electrochemical performances of the Nasicon SE-based Na-air batteries in ambient air as a fuel are superior than those of all-solid-state Li-air ($O_2$) batteries even without any liquid additives in air-electrodes[7,16,18], because of the higher redox potential of the $Na_2CO_3 \cdot xH_2O$ reactions (~3.4 V) and much smaller potential gap between the charge and discharge, which is originated from the same electrochemical reaction pathway in the charge/discharge. The substantial improvement in electrochemical performance in the Nasicon SE-based Na-air battery originates from the in-situ formed catholyte that can lead to the chemical reactions of the discharge products formed in 1$^{st}$ discharge with the ambient air, and then can activate the reversible electrochemical reactions of $Na_2CO_3 \cdot xH_2O$ (Fig. 4) in subsequent cycles. In consequence, these superior electrochemical performances demonstrates that the SE-based Na-air battery can be operated in ambient air with a wide range of humidity by exploiting reversible electrochemical reactions of sodium carbonates/hydroxides which can deliver high energy density with facile kinetic.

Furthermore, the carbonate reactions with high operating voltage and a low polarization can deliver higher theoretical energy density than that of $MO_2$ (M = Li or Na) (Fig. 4g and Supplementary Table S4) leading to small potential gap between charge and discharge than $M_2O_2$ or $M_2O$ (Supplementary Table S5). Compared to reported hybrid (aqueous + solid) electrolyte systems that have a large amount of $H_2O$, the Nasicon SE based Na-air battery is quite different because it exploits only the absorbed $H_2O$ obtained from the humidity of the ambient air, which can provide a very limited amount of $H_2O$, and thereby can have much higher volumetric and gravimetric energy density than the hybrid electrolyte systems. Moreover, the reversible electrochemical reactions involving $Na_2CO_3 \cdot xH_2O$ during the

charging and discharging cycles observed in the SE-based Na-air cell have never been previously reported in the hybrid electrolyte systems with flowing the air (not pure $O_2$) into the aqueous electrolyte[20,21,34]. This can be due to partly because of the absorption of a minimal amount of water that can facilitate the maintenance of a high local $CO_2$ concentration, which activates the reversible $Na_2CO_3 \cdot xH_2O$ reactions exclusively in the SE-based Na-air battery.

Utilizing ambient air without any additional devices has several advantages for practical use of metal-air batteries. Firstly, the cell design can be simplified because additional devices such as gas selective devices and gas tanks for storing purified gas are not necessary[6,35]. As a result, it will be helpful increase the gravimetric/volumetric energy densities of the metal-air batteries and step toward into the practical applications. Secondly, the Na-air cell in ambient air can achieve the lowest energy cost among various energy storage systems[36] because sodium is an earth-abundant material, the ambient air is free, and additional cost for preparing purified gas is not necessary. By using oxide-based solid electrolytes, the Na-air battery allows to use ambient air reversibly as a fuel, and enables to have chemical reactions between discharge products and the air that can lead to the formation of catholyte and can activate reversible electrochemical carbonate reactions, in contrast to previous approaches that try to suppress these chemical reactions.

In conclusion, we were able to employ ambient air as a fuel in a Nasicon SE-based Na-air cell capable of delivering reversible capacity by activating electrochemical reactions with carbonates and hydroxides, which are commonly believed to degrade the reversibility and induce high polarization in reported metal-air batteries. This counter-intuitive result can be explained by considering the role of an in-situ formed catholyte caused by the chemical reaction of the discharge products with $H_2O$, which activates the reversible electrochemical reaction of carbonates and facilitate its kinetics. Consequently, the Nasicon SE-based Na-air cell delivers high energy density due to high redox potential of the carbonate reaction as well as high energy efficiency due to low polarization by operating on the same electrochemical pathway during charge and discharge. This first demonstration of ambient air as a fuel in a SE-based Na-air battery provides steps toward developing beyond conventional metal-$O_2$ batteries and demonstrates the basis of employing reversible carbonate reactions to enable novel electrochemical energy platforms with solid electrolytes that can achieve much higher energy density than conventional batteries.

## Methods

### Preparation of the Nasicon SE based Na-air batteries

$Na_3Zr_2Si_2PO_{12}$ (Nasicon) solid electrolyte was synthesized by a solid-state reaction. A stoichiometric mixture of $Na_2CO_3$, $ZrO_2$, $SiO_2$, and $NH_4H_2PO_4$ (Na/Zr/Si/P = 3/2/2/1) was ball-milled for 24 h with 3, 5, and 10 mm $ZrO_2$ balls with ethanol solvent at 300 rpm. Then, the starting mixture was calcined at 1150 °C for 5 h under air to synthesize Nasicon phase, and the calcined sample was pulverized for 3 h by a planetary ball-mill at 500 rpm with 0.5 mm $ZrO_2$ balls with ethanol solvent in a $ZrO_2$ container. After that, the pulverized Nasicon was pressed in a mold with ~300 MPa. Lastly, the pressed sample was sintered at 1100 °C for 10 h under air atmosphere. The sintered Nasicon samples had a disk-shape with the density of ~3.0 g cm$^{-3}$, the diameter of ~12.20 mm, and the thickness of 1.00 mm.

To prepare the duplex structure of the Nasicon solid electrolyte, a porous Nasicon layer with 8 mm diameter was laminated on top of the dense Nasicon surface by screen printing process. The screen-printing process proceeded by printing a Nasicon paste with pore former on the sintered Nasicon, and then heating the screen-printed sample. First, a screen-printing solution was prepared by mixing alpha-terpineol as a solvent, ethylene cellulose as a binder, and di-ethylene glycol butyl ether as a pore former at a weight ratio of 60: 9: 31. Then, the Nasicon paste was prepared by mixing the pulverized Nasicon

powder (40 wt%), obtained by the planetary ball-mill process, and the screen-printing solution (60 wt%) using an agate mortar. The screen-printed paste on the sintered Nasicon was fired at 1100 °C for 5 h.

The air-electrodes were fabricated by forming with Ni nanoparticles inside the porous Nasicon of the duplex structure. The Ni nanoparticles were formed by using a solution-based infiltration process. A droplet (12 μl) of Ni solution (0.6 M $Ni(NO_3)_2 \cdot 6H_2O$ in distilled water) was injected into the porous Nasicon layer, and then the combination was heated at 700 °C for 10 min in ambient air to form NiO nanoparticles. After that, a porous Ni layer was stacked on the air-electrode as a current collector. To make the porous Ni layer, an Ni paste composed of Ni particles (40 wt%, <1 μm particle size) and the screen-printing solution (60 wt%) was mixed by using agate mortar, and then printed on the top of porous Nasicon layer of the duplex structure. The samples were heated to 300 °C for 10 min in ambient air, and then reheated to 700 °C for 10 min in a reducing atmosphere (95% Ar + 5% $H_2$) to transform residual NiO to Ni in the porous Nasicon layer and the current collector. A thin layer (~10 nm) of Au was deposited on the anode side of the dense Nasicon electrolyte in the duplex by using ion-coater (PS-1200, ParaOne) to make a homogeneous interface between Na metal anode and the dense Nasicon electrolyte.

### Cell assembly

The cell was composed of various hardware (Supplementary Fig. S4). The resulting SE-based Na-air cells were assembled in an Ar-filled glove box at $O_2$ and $H_2O$ contents <0.1 ppm by attaching a Na metal foil to the anode side of the cell (on the top of the Au layer). After assembling the cell, pure $O_2$ gas (99.995% purity) was blown toward the cathode side for 3 h to improve the integrity of the components of the cell. The flow rate of the gas was 0.3 mL min$^{-1}$.

For the hybrid electrolyte systems, the same cell structure as the Nasicon SE based Na-air cell was used but the cathode side of the cell is closed to the air unlike the Na-air battery, which has the air-electrode open to the air, to suppress the evaporation of the aqueous electrolytes during the test (Supplementary Fig. S16). The aqueous solutions of 0.6 M NaOH and 0.3 M $Na_2CO_3$ in distilled water were prepared at first. After that, the hybrid electrolyte cell was assembled in the Ar-filled glove box. After that, the aqueous solutions (1 mL) were injected to the cathode side of the battery, and then the cells were tested at room temperature.

### Electrochemical measurements

The assembled Na-air batteries were tested using a battery test system (Maccor 2200, Maccor, Inc.). The cell tests were performed at 25 °C in a constant-temperature-and-humidity chamber (TH-KE-025, JeioTech). The cells were discharged to 0.1, 0.2 or 0.8 mAh cm$^{-2}$ and charged up to 3.85 V at current densities of 0.02 for 1$^{st}$ cycle and 0.1 mA cm$^{-2}$ for subsequent cycles. For the hybrid electrolyte systems, the cells were charged to 0.2 mAh cm$^{-2}$, and then discharged to 0.2 mAh cm$^{-2}$ at a current density of 0.2 mA cm$^{-2}$. In rate capability tests, the cells were charged with the CCCV mode; after charging the cell up to 3.85 V at constant current and then the voltage was held at 3.85 V until the current reached <30% of the applied current. The cyclic voltammetry measurement was performed at a scan rate of 0.1 mV s$^{-1}$ under ambient air with 70% RH at 25 °C. Electrochemical procedures used for each cell are explained in the captions of the corresponding figures.

### Characterization techniques

Scanning electron microscopy (SEM) was performed to investigate the microstructure of the air-cathodes in the Na-air batteries with a field emission scanning electron microscopy (XL30s FEG, Philips Electron Optics B.V.) The reaction products of the air-electrodes were characterized using X-ray diffraction (XRD) measurements. The XRD spectra of the air-electrodes were recorded using diffractometer (D/MAX-2500/PC, Rigaku), using a step size of 0.02° at $10 \leq 2\theta \leq 70°$. The

synchrotron XRD pattern of the discharged air-electrode was also collected using beamline 9B at the Pohang Accelerator Laboratory (PAL) in Pohang, South Korea. Readings were recorded every 0.01° for 4 s. Confocal Raman spectroscopy (laser excitation wavelength $\lambda = 532$ nm; power = 1 mW; resolution = 1 μm; integration time = 10 s; confocal mode, Alpha 300 R, Witec, Ulm, Germany) was used to detect the discharge products in the air-electrode (Supplementary Figs. S8, 10, 11 13 and 17). The instrument is equipped with a microscope with a focal spot size of 500 nm. Real-time monitoring of gas evolution was achieved using a customized setup based on the VMP3 multichannel electrochemical workstation (Bio-Logic, France) and DEMS system as detailed in previous reports[37]. A lab-built gas-tight cell (Supplementary Fig. S9) was connected in line with an Ar gas (99.999%), multiport valve (VICI Valco, USA), and a quadrupole mass spectrometer (RGA200, Stanford Research Systems, USA). Throughout the galvanostatic charging process, Ar carrier gas periodically sweeps the evolved gases in the headspace, delivering them to the mass spectrometer chamber every five minutes. The recorded ionic current at $m/z = 44$ was used as $CO_2$ gas signal.

## Data availability
All data that support the findings of this study are presented in the Manuscript and Supplementary Information or are available from the corresponding author upon request.

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

## Acknowledgements

This study was supported by the Basic Science Research Program through the National Research Foundation of Korea (NRF) funded by the Ministry of Science, ICT & Future Planning (NRF-2017M3A7B8065394 and NRF-2019R1A2C2007933), and Brain Korea 21 PLUS project for Center for Creative Industrial Materials (no. F19SN25D1706). This work was supported by Korea Institute of Energy Technology Evaluation and Planning (KETEP) grant funded by the Korea government (MOTIE) (20215610100040, Development of 20 Wh seawater secondary battery unit cell). We are grateful to Dr. Kunjoong Kim in POSTECH for the insightful discussions on screen printing process.

## Author contributions

B. Kang and H. Park conceived the idea and design of the experiments, and in drafting the article for important intellectual content. H. Park carried out the experiments and data analysis and B. Kang supervised the experiments and analysis. H. Park and M. Kang performed ex-situ analyses. H. Park and D. Lee carried out CV analysis. M. Kang, J. Park, and S.J. Kang performed DEMS analysis. H. Park and B. Kang cowrote the paper.

## Competing interests

The authors declare no competing interests.
