## [Peer Review File · Nature Communications]

Activating reversible carbonate reactions in Nasicon solid electrolyte-based Na-air battery via in-situ formed catholyteReviewers' comments:

Reviewer #1 (Remarks to the Author):

Different from traditional Li/Na-O₂ batteries, the authors demonstrate a novel Na-air battery with both high discharge potential and high energy efficiency by taking the advantage of ambient air, including CO₂, O₂ and moisture, which shows higher energy output than most traditional Na-O₂/CO₂ batteries. Besides, considering this interesting working mechanism, it will benefit both energy storage and Carbon neutrality areas. However, the most important issue, i.e., the underlying mechanism of the high discharge plateau at 3.4 V, lacks sufficient supporting evidence although lots of experiments have been done. This high discharge potential may not be from the reaction that the authors assumed but from other reactions, for example, the Ni(OH)₂/NiOOH redox happens on the Ni current collector. Thus, this paper cannot be accepted at this stage.

Details can be found below:

Although the authors provide the evidence of Gibbs formation energy to show the potential of the induced reaction " $2\text{Na} + \text{CO}_2 (\text{g}) + 0.5\text{O}_2 (\text{g}) + \text{H}_2\text{O} (\text{g}) \rightarrow \text{Na}_2\text{CO}_3\text{H}_2\text{O} (\text{s})$ ", it still lacks strong evidence to prove this reaction is exactly the underlying mechanism of the high discharge plateau at 3.4 V. Most of the characterization experiments shown in this paper only proved that CO₂ has been involved in the whole reaction, but not this unique discharge plateau, especially considering most Na-O₂/CO₂ batteries shows a discharge voltage around 2.5V (Tong, Zizheng, et al. "Na-CO₂ battery with NASICON-structured solid-state electrolyte." *Nano Energy* 85 (2021): 105972.).

So, the authors must consider the possibility that other reactions may be induced in this battery system, which greatly promotes the battery voltage. For example, the battery utilized Ni as its current collector. However, the Ni metal is quite easy to be transformed into Ni(OH)₂/NiOOH in an alkaline environment (Trafela, Špela, et al. "Formation of a Ni (OH)₂/NiOOH active redox couple on nickel nanowires for formaldehyde detection in alkaline media." *Electrochimica Acta* 309 (2019): 346-353.), and the Ni(OH)₂/NiOOH redox can boost the battery voltage as a high discharge plateau of 3.2V vs Na/Na⁺ has been obtained by a rechargeable Na/Ni battery (Park, Seungyoung, et al. "Rechargeable Na/Ni batteries based on the Ni (OH)₂/NiOOH redox couple with high energy density and good cycling performance." *Journal of Materials Chemistry A* 7.4 (2019): 1564-1573.). Besides, a peak representing Ni-O around 550 cm⁻² in Raman spectra has been reported in this Na/Ni battery study, which can also be found in the Raman spectra of the reacted air-electrode shown in the Figure S12c of this paper. Thus, the authors should carefully check their evidence and find out what is going on inside this Na-air battery.

Thus, it is strongly recommended to do more electrochemical tests (such as CV scans) and in-situ/ex-situ characterization experiments (such as XPS, XRD, and Raman tests on different stages of charge/discharge processing) on the battery to further illustrate the authors' statement.

Reviewer #2 (Remarks to the Author):

This work reports the observation of reversible electrochemical formation of $\text{NaOH}\cdot\text{H}_2\text{O}$, which may open a revenue for energy storage. My main concern is the reaction mechanisms. Although detained electrochemical reaction with in-situ DEMS analysis is performed, the electrochemical reaction is still not convincing. It would be better if driving forces for all kinds of possible formation products can be included so that the thermodynamic data will be able to assist to understand the true story behind the observations.

Some grammatical error and typos should be corrected together with the study of mechanisms of electrochemical reaction.

Reviewer #3 (Remarks to the Author):

In this manuscript, the authors reported a reversible carbonate reaction in NASICON solid electrolyte ($\text{Na}_3\text{Zr}_2\text{Si}_2\text{PO}_{12}$), enabling the operation of sodium-air batteries under an ambient atmosphere. It is an inspiring work with new insight into the material design and reaction mechanism for reversibly achieving ambient air used as a fuel. The structure evolution and corresponding performance were detailed and discussed by experimental verification. The electrochemical performance was evaluated under diverse conditions, demonstrating high reversible electrochemical performance. However, this manuscript lacks a fine-grained characterization of critical mechanism, thus it is not recommended for publication in Nature Communications at present form. If the authors can address the following issues, I am glad to reconsider this study.

1. SEM shows a film-like morphology with discharge products covering the entire electrode. What is the composition of this film? Does it contain carbon deposits? Also, please give images with a larger scale to illustrate the uniformity.
2. Based on the reaction formula in the text, CO_2 , H_2O and O_2 are involved in the reaction process, but the content of CO_2 in the environment is extremely low. The article claims that the CO_2 produced by the reaction will be adsorbed near the electrode. How to prove it? How to ensure priority and sufficient CO_2 to participate in the reaction?
3. Whether it is possible to observe the proposed electrochemical reaction mechanism in the first and subsequent cycles more directly by in-situ Raman or in-situ XRD, etc.. Particularly, the authors should demonstrate the formation of $\text{Na}_2\text{CO}_3\cdot x\text{H}_2\text{O}$ at the high platform and low platform formation NaOH (Fig. S15) more concrete.
4. Why is the charging efficiency of the first lap significantly lower than other laps?
5. The effect of low humidity on the reaction is not complete, please supplement the relevant tests and characterizations under $\text{RH}\sim 20\%$.
6. Is it possible that hydrogen ions get electrons to produce gas during discharge? It can be verified by DEMS or gas chromatography.
7. Can a similar cathode reaction mechanism be realized in other metal-air battery systems?
8. The reaction from NaOH to $\text{Na}_2\text{CO}_3\cdot x\text{H}_2\text{O}$ during 1st discharge seems to be a chemical reaction rather than electrochemical reaction. How about the effect of shelving time on the electrochemical performance of this Na-air cell?

9. There are some spelling errors in the text, please review the full text and correct it.

For example,

Page 8, Line 187 ~ disappears

Page 13, Line 295 ~ On the contrary

Page 19, Line 433 ~ in the the Nasicon

The third and fourth steps in Figure 3e ~ During of 1st discharge, After of 1st discharge

.....

Response to comments

We thank all reviewers for their time spent to evaluate our manuscript and for their valuable comments. All the reviewers' concerns are addressed below in detail.

Reviewers' comments:

Reviewer #1 (Remarks to the Author):

Different from traditional Li/Na-O₂ batteries, the authors demonstrate a novel Na-air battery with both high discharge potential and high energy efficiency by taking the advantage of ambient air, including CO₂, O₂ and moisture, which shows higher energy output than most traditional Na-O₂/CO₂ batteries. Besides, considering this interesting working mechanism, it will benefit both energy storage and Carbon neutrality areas. However, the most important issue, i.e., the underlying mechanism of the high discharge plateau at 3.4 V, lacks sufficient supporting evidence although lots of experiments have been done. This high discharge potential may not be from the reaction that the authors assumed but from other reactions, for example, the Ni(OH)₂/NiOOH redox happens on the Ni current collector. Thus, this paper cannot be accepted at this stage.

Details can be found below:

Although the authors provide the evidence of Gibbs formation energy to show the potential of the induced reaction “ $2\text{Na} + \text{CO}_2 (\text{g}) + 0.5\text{O}_2 (\text{g}) + \text{H}_2\text{O} (\text{g}) \rightarrow \text{Na}_2\text{CO}_3\text{H}_2\text{O} (\text{s})$ ”, it still lacks strong evidence to prove this reaction is exactly the underlying mechanism of the high discharge plateau at 3.4 V. Most of the characterization experiments shown in this paper only proved that CO₂ has been involved in the whole reaction, but not this unique discharge plateau, especially considering most Na-O₂/CO₂ batteries shows a discharge voltage around 2.5V (Tong, Zizheng, et al. "Na-CO₂ battery with NASICON-structured solid-state electrolyte." Nano Energy 85 (2021): 105972.).

So, the authors must consider the possibility that other reactions may be induced in this battery

system, which greatly promotes the battery voltage. For example, the battery utilized Ni as its current collector. However, the Ni metal is quite easy to be transformed into Ni(OH)₂/NiOOH in an alkaline environment (Trafela, Špela, et al. "Formation of a Ni (OH)₂/NiOOH active redox couple on nickel nanowires for formaldehyde detection in alkaline media." *Electrochimica Acta* 309 (2019): 346-353.), and the Ni(OH)₂/NiOOH redox can boost the battery voltage as a high discharge plateau of 3.2V vs Na/Na⁺ has been obtained by a rechargeable Na/Ni battery (Park, Seungyoung, et al. "Rechargeable Na/Ni batteries based on the Ni (OH)₂/NiOOH redox couple with high energy density and good cycling performance." *Journal of Materials Chemistry A* 7.4 (2019): 1564-1573.). Besides, a peak representing Ni-O around 550 cm⁻² in Raman spectra has been reported in this Na/Ni battery study, which can also be found in the Raman spectra of the reacted air-electrode shown in the Figure S12c of this paper. Thus, the authors should carefully check their evidence and find out what is going on inside this Na-air battery.

Thus, it is strongly recommended to do more electrochemical tests (such as CV scans) and in-situ/ex-situ characterization experiments (such as XPS, XRD, and Raman tests on different stages of charge/discharge processing) on the battery to further illustrate the authors' statement.

→ We thank the reviewer for the constructive comments. We would like to emphasize the fact that the 3.4 V plateau was barely increased by repeating cycles in the hybrid Na-air cell with NaOH(aq) solution (Fig. S14) even though Ni particles were exposed to significant amount of OH⁻ ions. It indicates that the conversion from Ni to Ni(OH)₂/NiOOH is very difficult. The reviewer commented that Ni metal is quite easy to transformed into Ni(OH)₂/NiOOH in alkaline solution based on the previous study ("Formation of a Ni(OH)₂/NiOOH active redox couple on nickel nanowires for formaldehyde detection in alkaline media." *Electrochimica Acta* 309 (2019): 346-353.). However, the authors of the study clearly demonstrated that the untreated Ni (nanowire) barely transforms into Ni(OH)₂/NiOOH due to a presence of a native oxide on the surface such as NiO. Similarly, the Ni current collector in our cell was covered by Ni oxides as shown in XPS data (Fig. R1). Considering that Ni metal, not Ni oxides was strongly observed by XRD (Fig. S12), it can conclude that a thin Ni oxides surface layer covers Ni particles in the SE-based Na-air cell. As a result, the electrochemical reaction of Ni is not dominantly occurred in our SE-based Na-air cell.

Furthermore, we confirmed the effect of Ni content in the air-electrode on the voltage plateau. By using the Ni infiltration process as explained experimental section in the manuscript, we prepared additional cells that has increased amount of Ni in the air-electrode compared to the pristine cell (Fig. R2). Even though the Ni content in the air-electrode become double or triple, the ~ 3.4 V plateau doesn't increase as the Ni content increases, and thereby the ~ 3.4 V voltage plateau is barely increased according to the amount of Ni. This clearly indicates that the Ni does not participate in the electrochemical reaction.

Combining these results, we conclude that Ni particles was barely transformed into $\text{Ni}(\text{OH})_2/\text{NiOOH}$ during cycling and thereby the high discharge voltage at ~ 3.4 V is not from $\text{Ni}(\text{OH})_2/\text{NiOOH}$.

We added this explanation in the manuscript and Supplementary information.

Figure R1. XPS of Ni current collector of the pristine SE-based Na-air cell; Ni oxides as well as Ni metal were observed.

Figure R2. a-c, Comparison of voltage profiles of the SE-based Na-air cell. Ni infiltration process performed for (a) pristine cell (2 times), (b) the cell with 4 times Ni and (c) the cell with 6 times Ni, respectively.

Reviewer #2 (Remarks to the Author):

This work reports the observation of reversible electrochemical formation of $\text{NaOH}\cdot\text{H}_2\text{O}$, which may open a revenue for energy storage. My main concern is the reaction mechanisms. Although detailed electrochemical reaction with *in-situ* DEMS analysis is performed, the electrochemical reaction is still not convincing. It would be better if driving forces for all kinds of possible formation products can be included so that the thermodynamic data will be able to assist to understand the true story behind the observations.

Some grammatical error and typos should be corrected together with the study of mechanisms of electrochemical reaction.

→ We appreciate valuable comments. We would like to emphasize that we already introduced all kinds of possible electrochemical/chemical reactions that can be occurred in the Na-air system (Table S1 and S2). We cross-checked the decomposition of the discharge products by XRD, Raman, and *in-situ* DEMS. However, as we already mentioned in supplementary information, the DEMS instrument can't distinguish the variance of each gas in the mixed gases (O_2 , CO_2 , H_2O) during discharge. Therefore, the *in-situ* DEMS study during discharge wasn't proceeded.

Instead, we performed *ex-situ* XRD and Raman analysis to observe the proposed electrochemical reaction mechanism (Figure R3). When the SE-based Na-air cell was discharged to 2.8 V, the formation of the discharge products ($\text{Na}_2\text{CO}_3\cdot x\text{H}_2\text{O}$) was observed by Raman, not XRD. It might be originated by a small quantity of the discharge products. The CO_3^{2-} band of Na_2CO_3 (1080 cm^{-1}) and OH bands of H_2O ($3200\text{-}3500\text{ cm}^{-1}$) was detected in the cell discharged to 2.8 V. When the discharge process is further proceeded at low potential, the intensity of XRD peaks corresponding to NaOH increased compared to those of $\text{Na}_2\text{CO}_3\cdot x\text{H}_2\text{O}$ and Nasicon. The similar result was also observed in the *ex-situ* Raman analysis of the hybrid Na-air cell (Fig. S10 in Supplementary information). Combining these results, we can conclude that the $\text{Na}_2\text{CO}_3\cdot x\text{H}_2\text{O}$ and NaOH is electrochemically formed at high ($\sim 3.4\text{ V}$) and low ($\sim 2.0\text{ V}$) potential, respectively.

We added this result in the Supplementary information (Supplementary Fig. S16).

In addition, we revised typos in the manuscript and the Supplementary information.

Figure R3. a, Voltage profiles of the SE-based Na-air cells discharged to 0.8 and 1.5 mAh cm⁻². **b,** *Ex-situ* XRD patterns of the air electrode with different discharge cutoff conditions: 2.8 V, 0.8 and 1.5 mAh cm⁻². We note that the cell was pre-activated for 10 cycles by discharging to 0.2 mAh cm⁻² and charging up to 3.85 V at a current density of 0.1 mAh cm⁻². **c,** Raman spectra of the air-electrode in the cell discharged to 2.8 V.

Reviewer #3 (Remarks to the Author):

In this manuscript, the authors reported a reversible carbonate reaction in NASICON solid electrolyte ($\text{Na}_3\text{Zr}_2\text{Si}_2\text{PO}_{12}$), enabling the operation of sodium-air batteries under an ambient atmosphere. It is an inspiring work with new insight into the material design and reaction mechanism for reversibly achieving ambient air used as a fuel. The structure evolution and corresponding performance were detailed and discussed by experimental verification. The electrochemical performance was evaluated under diverse conditions, demonstrating high reversible electrochemical performance. However, this manuscript lacks a fine-grained characterization of critical mechanism, thus it is not recommended for publication in Nature Communications at present form. If the authors can address the following issues, I am glad to reconsider this study.

1. SEM shows a film-like morphology with discharge products covering the entire electrode. What is the composition of this film? Does it contain carbon deposits? Also, please give images with a larger scale to illustrate the uniformity.

→ The cross-sectional view of the air-electrode in the discharged and charged cells ($0.8 \text{ mAh/cm}^2 - 3.85 \text{ V}$) is included in Fig. 3b, and the SEM images with a larger scale are added in Fig. R4. As shown in the SEM images, the discharge products uniformly cover entirely the air-electrode and the products are decomposed by charge (Fig. R4a-b). Carbon element in the discharged cell was detected by EDS mapping implying the presence of $\text{Na}_2\text{CO}_3 \cdot x\text{H}_2\text{O}$ (Fig. R4c-d). However, peaks of the other elements (Zr, P, Si) contained in Nasicon were also observed so that it is hard to analyze the composition of the film using SEM/EDS. Instead, the composition of the film was already analyzed using XRD, Raman as shown in the manuscript and Supporting information.

Figure R4. a-b, The cross-sectional view of the air-electrode after (a) discharging and (b) charging the cell ($0.8 \text{ mAh cm}^{-2} - 3.85 \text{ V}$). **c**, The SEM image of the discharge products with a magnification. **d**, EDS analysis of the discharged products based on Fig. R4c.

2. Based on the reaction formula in the text, CO_2 , H_2O and O_2 are involved in the reaction process, but the content of CO_2 in the environment is extremely low. The article claims that the CO_2 produced by the reaction will be adsorbed near the electrode. How to prove it? How to ensure priority and sufficient CO_2 to participate in the reaction?

→ Figure S9 shows the *in-situ* DEMS analysis during 1st charge. Considering the applied current (0.02 mA cm^{-2}) and the Na_2CO_3 reaction (equation 4 in the manuscript) and the Faraday's law of electrolysis, the theoretical evolution rate of CO_2 gas during charge is $3.1 \text{ nmol min}^{-1}$, but the observed rate was $\sim 1 \text{ nmol min}^{-1}$. This discrepancy would be originated from the dissolution of CO_2 in the absorbed water of the air-electrode. As the reviewer said, CO_2 in ambient air is extremely low. Therefore, repeating charging/discharging processes as already observed in the study would be needed to increase CO_2 concentration in the air-electrode by the similar amount of O_2 concentration in ambient air.

3. Whether it is possible to observe the proposed electrochemical reaction mechanism in the first and subsequent cycles more directly by in-situ Raman or in-situ XRD, etc.. Particularly, the authors should demonstrate the formation of $\text{Na}_2\text{CO}_3 \cdot x\text{H}_2\text{O}$ at the high platform and low platform formation NaOH (Fig. S15) more concrete.

→ We performed *ex-situ* XRD and Raman analysis to observe the proposed electrochemical reaction mechanism (Figure R5). When the SE-based Na-air cell was discharged to 2.8 V, the formation of the discharge products ($\text{Na}_2\text{CO}_3 \cdot x\text{H}_2\text{O}$) was observed by Raman, not XRD. It might be originated by a small quantity of the discharge products. The CO_3^{2-} band of Na_2CO_3 (1080 cm^{-1}) and OH bands of H_2O ($3200\text{-}3500 \text{ cm}^{-1}$) was detected in the cell discharged to 2.8 V. When the discharge process is further proceeded at low potential, the intensity of XRD peaks corresponding to NaOH increased compared to those of $\text{Na}_2\text{CO}_3 \cdot x\text{H}_2\text{O}$ and Nasicon. The similar result was also observed in the *ex-situ* Raman analysis of the hybrid Na-air cell (Fig. S10 in Supplementary information). Combining these results, we can conclude that the $\text{Na}_2\text{CO}_3 \cdot x\text{H}_2\text{O}$ and NaOH is electrochemically formed at high ($\sim 3.4 \text{ V}$) and low ($\sim 2.0 \text{ V}$) potential, respectively.

We added this result in the Supplementary information (Supplementary Fig. S16).

Figure R5. a, Voltage profiles of the SE-based Na-air cells discharged to 0.8 and 1.5 mAh cm⁻². **b,** *Ex-situ* XRD patterns of the air electrode with different discharge cutoff conditions: 2.8 V, 0.8 and 1.5 mAh cm⁻². We note that the cell was pre-activated for 10 cycles by discharging to 0.2 mAh cm⁻² and charging up to 3.85 V at a current density of 0.1 mAh cm⁻². **c,** Raman spectra of the air-electrode in the cell discharged to 2.8 V.

4. Why is the charging efficiency of the first lap significantly lower than other laps?

→ Small amount of H₂O compared to that of discharge product could be absorbed at 1st cycle, and it can cause low coulombic efficiency of the SE-based Na-air cell. As shown in Fig. S18b and S19b, the overpotential during charge increases by lowering the relative humidity of air. Taking this observation into account, it can be demonstrated that the *in-situ* formed catholyte at 1st cycle isn't sufficiently formed to enhance ionic conduction through the air-electrode or inside air-electrode.

5. The effect of low humidity on the reaction is not complete, please supplement the relevant tests and characterizations under RH~20%.

→ The SE-based Na-air cell was operated in ambient air (Figure R6). We note that the cell was pre-cycled for 10 cycles in air with 70% RH at 25°C, and the current density was 0.1 mA cm⁻². The temperature of ambient air was maintained around 24°C, however, the relative humidity fluctuated by 20%.

As shown in Figure R6a, the variance of coulombic efficiency was similar to that of the relative humidity. It implies that the amount of H₂O in the air-electrode changed by the humidity strongly affects the kinetics. At 40th cycle, the relative humidity decreased from 70% to ~35%. The 2nd discharge plateau was decreased by the ~3.4 V plateau was maintained. However, when the relative humidity decreased to 21% (65th cycle), the capacity of the ~3.4 V plateau during discharge was reduced and the two plateaus were observed clearly during charge (Figure R6b). It can support that lowering the absorbed H₂O under low humidity can affect the chemical reaction between NaOH(discharge product) and CO₂ and the kinetics.

We added this result in the Supplementary information.

Figure R6. The electrochemical performance of the SE-based Na-air cell operated in ambient air after pre-cycling the cell in air with 70% RH at 25°C. a, Variances of the cell performance and ambient air conditions during cycle test. **b,** Voltage profiles of the SE-based Na-‘ambient air’ cell in different relative humidity conditions. The cell was discharged to 0.2 mAh cm⁻², and charged to 3.85 V at a current density of 0.1 mA cm⁻²..

6. Is it possible that hydrogen ions get electrons to produce gas during discharge? It can be verified by DEMS or gas chromatography.

→ As we explained in the Supplementary information (section 8.2), the DEMS instrument what we used can only analyze the variance in the entire amount of mixed gas, not that of each

gas. Therefore, it was very difficult to analyze evolved H₂ gas flowing mixed gas (O₂, CO₂, H₂O) during discharge. The reviewer might be suspicious that the ~2.5 V plateau originated by the reaction of $2\text{Na} + 2\text{H}^+ + \rightarrow 2\text{Na}^+ + \text{H}_2$ suggested by C. Kim *et al.* (*iScience* **9**, 278–285 (2018)).

To activate H₂ evolution reaction during discharge, considerable amount of H⁺ ions must be generated. One way is to decompose H₂O ($2\text{H}_2\text{O} \rightarrow \text{O}_2 + 4\text{H}^+ + 4\text{e}^-$, $E^0 = 1.23 \text{ V S.H.E}$) during charge, and another way is to induce CO₂ hydrolysis as C. Kim *et al.* reported. However, we controlled charge cutoff potential (3.85 V) to avoid H₂O decomposition (Fig. S12 in the Supplementary information). Also, our Na-air cell isn't suitable to induce CO₂ hydrolysis continuously because CO₂ in the absorbed H₂O is firstly consumed at ~3.4 V and CO₂ concentration in air is negligible. As a result, we don't think that the discharge reactions are involved with the H₂ evolution reaction for above reasons.

7. Can a similar cathode reaction mechanism be realized in other metal-air battery systems?

→ Different kinds of metal (M = Li, Na, Zn etc.)-air batteries can be operated based on formation/decomposition of MOH/M₂CO₃. However, it is known that the hygroscopicity of MOH/M₂CO₃ might be highest if M = Na. Therefore, the reaction mechanism would not be facile to be realized in other metal-air batteries.

8. The reaction from NaOH to Na₂CO₃·xH₂O during 1st discharge seems to be a chemical reaction rather than electrochemical reaction. How *about the effect of shelving time* on the electrochemical performance of this Na-air cell?

→ Voltage profiles of the SE-based Na-air cells were compared by increasing rest time after 1st discharge (Figure R7). The overpotential during 1st discharge of these cells was almost identical, however, after 1st charge, the cell with longer rest time exhibited higher overpotential. In addition, it is noteworthy that the increasing rate of the ~3.4 V plateau was lowered by extending the rest time more than 24 hours.

If all NaOH was transformed into Na₂CO₃·xH₂O, the absorbed H₂O in the air-electrode would be diminished because hygroscopicity of Na₂CO₃·xH₂O is lower than NaOH. It would result in the increase of the overpotential of the cell and the decrease in CO₂ concentration at the air-electrode.

In this study, we focus on the activation mechanism of the reversible $\text{Na}_2\text{CO}_3 \cdot x\text{H}_2\text{O}$ reactions in ambient air. Therefore, we didn't add experimental result of the shelving time of the SE-based Na-air cell.

Figure R7. a-c, Voltage profiles of the SE-based Na-air cells: (a) the voltage profiles of 1st charge after 1st discharge with different rest times, (b) voltage profiles of 2nd charge after 2nd discharge with different rest times, and (c) voltage profiles of 3rd discharge.

9. There are some spelling errors in the text, please review the full text and correct it.

For example,

Page 8, Line 187 ~ disappears

Page 13, Line 295 ~ On the contrary

Page 19, Line 433 ~ in the the Nasicon

The third and fourth steps in Figure 3e ~ During of 1st discharge, After of 1st discharge

.....

We revised typos in the manuscript and the supplementary information.

REVIEWER COMMENTS

Reviewer #1 (Remarks to the Author):

It is appreciated that the authors addressed the impossibility of Ni(OH)₂/NiOOH reactions well. However, the major problem of this article is that the characterization of the battery running mechanism is insufficient. Although excellent performance has been obtained, the supporting evidence is not strong, especially considering the reactions pointed out by the authors. The cathodic reactions listed by the authors are quite novel compared with widely utilized ORR, but the role of O₂ and CO₂ should be further examined carefully. In my opinion, such high battery performance should be combined with convincing evidence for publication. Otherwise, it may mislead the following studies.

The detailed problems were listed below:

1. As the Nasion electrolyte is widely used, the anodic reaction is quite clear. However, the cathodic reactions are quite confusing. Normally, the ORR reaction happens in the Nasion-based Na-air battery (<https://doi.org/10.1016/j.elecom.2015.10.004>) and shows a full cell voltage of around 2.8 V, which depends on the environmental pH. However, a quite high discharge voltage over 3 V was achieved by the authors, which is concluded due to the reversible Na₂CO₃·xH₂O (x = 0 or 1) reactions rather than the ORR. The authors should further check the reactions that happen on the cathode and carefully study the role of CO₂. As the reactions related to CO₂ seem to provide no electron, why it can boost the battery voltage? Although the DEMS measurement did show a lower amount of CO₂, it cannot prove its role in the electrochemical reaction. For example, some researchers also induced CO₂ into Nasion-based Na battery systems (<https://doi.org/10.1016/j.isci.2018.10.027>), which only plays a role in dissolution/production. Thus, the authors should carefully check the cathodic reactions and explain how CO₂ supports the reaction. Supplementary 8-7 and 8-8 provided some evidence but not enough. Electrochemical tests by a 3-electrode system shall be conducted separately with different electrolytes, i.e., this in-situ formed catholyte, NaHCO₃, and NaOH, and consider more factors, including pH, gases, and humidity.
2. The author stated that the reversible formation/decomposition of NaOH/Na₂CO₃ happens, however, they think the equation (4) and (5) are reversible during the battery runtime. So, what is the role of Na during the battery runtime? It is well known that Na is hard to be formed in an aqueous environment.
3. The electrochemical characterization has not been provided sufficiently. For example, CV curves are not mentioned in this article.

Reviewer #2 (Remarks to the Author):

The revised manuscript fully considered my concern and therefore I recommend for consideration of publication.

Reviewer #3 (Remarks to the Author):

After reviewing the revised version, I think that the authors have addressed the issues raised by the reviewers, and the quality of the manuscript has been significantly improved.

However, the main mechanisms proposed in this paper still lack sufficient and in-depth experimental evidence. Here are some suggestions for the authors.

1. The air electrode in the circulation process involves both electrochemical and chemical reactions from NaOH to Na₂CO₃. The ex-situ reaction cannot track species changes in real time, which may lead to inaccurate experimental results. Firstly, it is stated that Na₂O₂ is generated initially, which immediately combines with moist air to form NaOH, suggesting that the mechanism outlined in this study may be incomplete. Secondly, the sample may react with air during the early stages of the testing procedure. Therefore, in situ characterization (such as in situ Raman and in situ XRD) is required.

2. It is suggested that the author conduct XPS, SEM and XRD experiments on the recycled air cathode to determine whether Ni is transformed into other species, such as Ni(OH)₂. It is well known that Ni(OH)₂ with nanosheet structure is also an excellent electrocatalyst, which can greatly reduce the polarization phenomenon in the charge discharge curve.

3. In Ref. 19, a humid environment is also formed at the air cathode, but it is clearly pointed out that CO₂ causes the appearance of Na₂CO₃ and negatively affects the battery performance. Please explain the difference between this work and Ref. 19, as well as why Na₂CO₃ can reversibly transform in this article.

We appreciate all reviewers for their time spent to evaluate our manuscript and for their valuable comments. All the reviewers' concerns are addressed below in detail.

Reviewer #1 (Remarks to the Author):

It is appreciated that the authors addressed the impossibility of Ni(OH)₂/NiOOH reactions well. However, the major problem of this article is that the characterization of the battery running mechanism is insufficient. Although excellent performance has been obtained, the supporting evidence is not strong, especially considering the reactions pointed out by the authors. The cathodic reactions listed by the authors are quite novel compared with widely utilized ORR, but the role of O₂ and CO₂ should be further examined carefully. In my opinion, such high battery performance should be combined with convincing evidence for publication. Otherwise, it may mislead the following studies.

The detailed problems were listed below:

1. As the Nasicon electrolyte is widely used, the anodic reaction is quite clear. However, the cathodic reactions are quite confusing. Normally, the ORR reaction happens in the Nasicon-based Na-air battery (<https://doi.org/10.1016/j.elecom.2015.10.004>) and shows a full cell voltage of around 2.8 V, which depends on the environmental pH. However, a quite high discharge voltage over 3 V was achieved by the authors, which is concluded due to the reversible Na₂CO₃·xH₂O (x = 0 or 1) reactions rather than the ORR. The authors should further check the reactions that happen on the cathode and carefully study the role of CO₂. As the reactions related to CO₂ seem to provide no electron, why it can boost the battery voltage? Although the DEMS measurement did show a lower amount of CO₂, it cannot prove its role in the electrochemical reaction. For example, some researchers also induced CO₂ into Nasicon-based Na battery systems (<https://doi.org/10.1016/j.isci.2018.10.027>), which only plays a role in dissolution/production. Thus, the authors should carefully check the cathodic reactions and explain how CO₂ supports the reaction. Supplementary 8-7 and 8-8 provided some evidence but not enough. Electrochemical tests by a 3-electrode system shall be conducted separately with different electrolytes, i.e., this in-situ formed catholyte, NaHCO₃, and NaOH, and consider more factors, including pH, gases, and humidity.

→ No matter what CO₂ can provide electrons or not, Gibb's formation energies of sodium carbonates are lower than that of hydroxides and oxides. Therefore, redox potential can be elevated when the main redox reaction is transformed from NaOH to Na₂CO₃·xH₂O reaction.

We already showed that the SE-based Na-air cell with H₂O and O₂ without CO₂ did not have any ~3.4 V reaction (Fig. S16d). This clearly indicates that this 3.4 V redox reaction is not related to NaOH. In order to confirm pH effect, we also prepared a hybrid Na-air cell with 0.01M (pH ~ 11.2) and 2.0M (pH ~ 12.3) Na₂CO₃ aqueous solution (Fig. R1). If the ~3.4 V discharge plateau was driven by the ORR reaction under low pH environment, higher capacity would be exhibited in the 0.01M Na₂CO₃ cell rather than the 2.0M cell, yet the result showed the contrary. We note that 1 ml of the aqueous solution was used in the hybrid Na-air cells so that theoretical capacity of the 0.01M Na₂CO₃ solutions was 1.07 mAh cm⁻², which is higher than the capacity cut-off condition during discharge (0.2 mAh cm⁻²). The result implies that the

3.4 V plateau is not strongly influenced by pH of the *in-situ* formed catholyte.

Lastly, instead of the 3-electrode test, we added *in-situ* Raman analysis to support previously performed analysis (GITT, XRD, Raman, DEMS, etc.). We prepared new cell hardware for *in-situ* Raman measurement (Fig. R2a), and the top of the Ni current collector was observed. We note that the high cell polarization was observed due to the uncontrolled and low humidity in the environment (Fig. R2b). The cell was activated for 5 cycles in air with 70% RH before the *in-situ* Raman test. It is noteworthy that the peak at $\sim 1080\text{ cm}^{-1}$ corresponding to Na_2CO_3 grows during discharge (1~4 in Fig. R2c) is exhibited and then disappears by charging the cell (5~8 in Fig. R2d). The result of the *in-situ* Raman measurement demonstrates that $\text{Na}_2\text{CO}_3 \cdot x\text{H}_2\text{O}$ is electrochemically formed during discharge.

Differing from the *ex-situ* Raman experiment results, we did not observe any changes in the NaOH peak during the *in-situ* Raman experiment. This might be attributed to the fact that, unlike the *ex-situ* Raman measurement, which observed the entire electrode, the *in-situ* Raman experiment focused solely on the upper part of the electrode.

We added the *in-situ* Raman analysis in the revised manuscript and Supplementary Information.

Figure R1. Voltage profiles of hybrid (Na_2CO_3 (aq) + Nasicon) Na-air batteries with different PH

Figure R2. *in-situ* Raman analysis of the SE-based Na-air cell. **a**, design of *in-situ* Raman cells (inset: optical image showing upper-side of the air-electrode), **b**, Voltage profile of the SE-based Na-air cell during *in-situ* Raman test (current density: 0.2 mA cm⁻²), **c-d**, *in-situ* Raman spectra during (c) discharge and (d) charge.

2. The author stated that the reversible formation/decomposition of NaOH/Na₂CO₃ happens, however, they think the equation (4) and (5) are reversible during the battery runtime. So, what is the role of Na during the battery runtime? It is well known that Na is hard to be formed in an aqueous environment.

→ As shown in Fig. S4 and S16a, the anode side (Na metal) of the SE-based Na-air cell is firmly sealed by sealant so that the inert environment (Ar-filled), not aqueous environment, is maintained during the battery runtime. Therefore, the anodic reaction of the SE-based Na-air cell is as follows (equation R1):

3. The electrochemical characterization has not been provided sufficiently. For example, CV curves are not mentioned in this article.

→ The dQ/dV plot as shown in Fig. 2 of the manuscript can also provide similar information

on electrochemical reactions with CV curves. In addition, we added *in-situ* Raman analysis (Fig. R2) to support the electrochemical analysis (GITT, XRD, Raman, DEMS, etc.).

Reviewer #3 (Remarks to the Author):

After reviewing the revised version, I think that the authors have addressed the issues raised by the reviewers, and the quality of the manuscript has been significantly improved. However, the main mechanisms proposed in this paper still lack sufficient and in-depth experimental evidence. Here are some suggestions for the authors.

1. The air electrode in the circulation process involves both electrochemical and chemical reactions from NaOH to Na₂CO₃. The ex-situ reaction cannot track species changes in real time, which may lead to inaccurate experimental results. Firstly, it is stated that Na₂O₂ is generated initially, which immediately combines with moist air to form NaOH, suggesting that the mechanism outlined in this study may be incomplete. Secondly, the sample may react with air during the early stages of the testing procedure. Therefore, in situ characterization (such as in situ Raman and in situ XRD) is required.

→ As the reviewer commented, some previous studies, investigated on metal-air(O₂) cells, reported that sodium oxides (Na_xO_y) ‘electrochemically’ generated, and then, sodium hydroxide (NaOH) is ‘chemically’ formed by combining with moist air.[<https://doi.org/10.1002/adfm.202011151>] However, in our study, the discharge plateau during 1st cycle appears at ~ 2.7 V, which is much higher than thermodynamic potential of Na_xO_y (equations R2-R4) [<https://doi.org/10.1038/nmat3486>]:

Considering the above reactions, it is impossible that sodium oxides is electrochemically generated at higher voltage than their thermodynamic redox potential. Therefore, we ruled out the possibility of the electrochemical formation of sodium oxides during cycling the SE-based Na-air cell.

In addition, as the reviewer suggested, we added *in-situ* Raman analysis to support previously performed analysis (GITT, XRD, Raman, DEMS, etc.). We prepared new cell hardware for *in-situ* Raman measurement (Fig. R3a), and the top of the Ni current collector was observed. We note that the high cell polarization was observed due to the uncontrolled and low humidity in the environment (Fig. R3b). The cell was activated for 5 cycles in air with 70% RH before the *in-situ* Raman measurement. It is noteworthy that the peak at ~1080 cm⁻¹ corresponding to Na₂CO₃ grows during discharge (1~4 in Fig. R3c) is exhibited and then disappears by charging the cell (5 ~8 in Fig. R3d). The result of the *in-situ* measurement demonstrates that Na₂CO₃·xH₂O is electrochemically formed during discharge.

Differing from the *ex-situ* Raman analysis, we did not observe any changes in the NaOH peak during the *in-situ* Raman experiment. This might be attributed to the fact that, unlike the *ex-situ* Raman measurement, which observed the entire electrode, the *in-situ* Raman experiment focused solely on the upper part of the electrode.

We added the *in-situ* Raman analysis in the revised manuscript and Supplementary Information.

Figure R3. *in-situ* Raman analysis of the SE-based Na-air cell. **a**, design of *in-situ* Raman cells (inset: optical image showing upper-side of the air-electrode), **b**, Voltage profile of the SE-based Na-air cell during *in-situ* Raman test (current density: 0.2 mA cm⁻²), **c-d**, *in-situ* Raman spectra during (c) discharge and (d) charge.

2. It is suggested that the author conduct XPS, SEM and XRD experiments on the recycled air cathode to determine whether Ni is transformed into other species, such as Ni(OH)₂. It is well known that Ni(OH)₂ with nanosheet structure is also an excellent electrocatalyst, which can greatly reduce the polarization phenomenon in the charge discharge curve.

→ We thank the reviewer for the constructive comments. It is well known that Ni metal is quite easy to be transformed into Ni(OH)₂/NiOOH in an alkaline environment. [https://doi.org/10.1016/j.electacta.2019.04.060] However, we would like to emphasize the fact that the 3.4 V plateau was barely increased by repeating cycles in the hybrid Na-air cell with NaOH(aq) solution (Fig. S16) even though Ni particles were exposed to significant amount of OH⁻ ions. It indicates that the conversion from Ni to Ni(OH)₂/NiOOH is very difficult.

Taking a closer look at the previous study [https://doi.org/10.1016/j.electacta.2019.04.060], the authors of the study clearly demonstrated that the untreated Ni (nanowire) barely transforms into Ni(OH)₂/NiOOH due to a presence of a native oxide on the surface such as NiO. Similarly,

the Ni current collector in our cell was covered by Ni oxides as shown in XPS data (Fig. R4). Considering that Ni metal, not Ni hydroxides was strongly observed by XRD (Fig. S2) even though charging the cell, it can be concluded that a thin Ni oxides surface layer covers Ni particles in the SE-based Na-air cell. As a result, the electrochemical reaction of Ni is not dominantly occurred in our SE-based Na-air cell.

Figure R4. XPS of Ni current collector of the pristine SE-based Na-air cell; Ni oxides as well as Ni metal were observed.

3. In Ref. 19, a humid environment is also formed at the air cathode, but it is clearly pointed out that CO₂ causes the appearance of Na₂CO₃ and negatively affects the battery performance. Please explain the difference between this work and Ref. 19, as well as why Na₂CO₃ can reversibly transform in this article.

→ In ref. 19, a quasi-solid-state Na-air battery with gel cathode, composed of single-walled carbon nanotubes and ionic liquids (1-ethyl-3-methylimidazolium), is reported. In the study, the authors pointed out the gel is a ‘hydrophobic’ material. It implies that the absorption of H₂O on the air-electrode can be constrained by the presence of the hydrophobic material between moist air and the air-electrode. However, in our study, moisture in ambient air can be absorbed directly in the air-electrode, a large amount of H₂O can be accumulated in the air-electrode.

The amount of the absorbed H₂O results in the difference between the Na-air batteries. In the quasi-solid-state battery reported in Ref. 19, Na₂O₂ or NaO₂ is electrochemically generated at the early state of cycling due to lack of H₂O concentration, and then, NaOH is formed by chemical reaction between moisture in air and sodium oxides. However, as the aqueous layer is gradually formed by cycling the cell, NaOH is electrochemically formed, and then, NaOH is transformed into Na₂CO₃ by chemical reaction between NaOH and CO₂ in air. On the other hand, in case of the SE-based Na-air battery (our study), NaOH is electrochemically formed at the early stage of cycling due to high concentration of O₂ and H₂O, and then, Na₂CO₃·xH₂O is also formed/decomposed just after 1st cycle.

We are very cautious about comparing redox mechanisms between Ref. 19 and our study because of following reasons. Firstly, we couldn’t find CO₂ solubility of the ionic liquid yet. Also, the voltage profiles of the quasi-solid-state battery reported in Ref. 19 are always similar

whatever the discharge product is ($\text{Na}_2\text{O}_2 \cdot 2\text{H}_2\text{O}$), NaOH , and Na_2CO_3). However, considering $\text{Na}_2\text{CO}_3 \cdot x\text{H}_2\text{O}$ is soluble in H_2O , higher amounts of $\text{Na}_2\text{CO}_3 \cdot x\text{H}_2\text{O}$ can be dissolved and ionized in the SE-based Na-air cell (our study) compared to the quasi-solid-state Na-air cell (Ref. 19). It is noteworthy that the aqueous $\text{Na}_2\text{CO}_3 \cdot x\text{H}_2\text{O}$ solution is a Na-ion conductor that can support ion conduction of the air-electrode. We suppose that it might be the reason why Na_2CO_3 affects the battery performance differently depending on the type of the air-electrode.

REVIEWER COMMENTS

Reviewer #1 (Remarks to the Author):

In this version, the authors have provided evidence that addresses various concerns. From the given equations, it seems that the authors have developed a well-designed battery structure to achieve a CO₂ and H₂O assisted Na-O₂ battery system, in which the ORR can be considered to occur in an aqueous environment. So the ORR gets a higher potential compared with other organic electrolyte based Na-O₂ battery systems. Therefore, the authors should enhance the description of ORR in the manuscript to make it more reader-friendly before acceptance. Additionally, there are some points that need further revision.

1. In equations (1), (2), and (4), Na is shown as a reactant related to the formation of NaOH, as it only reacts at the anode. The authors should replace these equations with half-equations.
2. CV curves are still necessary, as they will not only display reduction reactions but also oxidation reactions. Both are important for rechargeable batteries.

Reviewer #3 (Remarks to the Author):

The authors have revised the manuscript carefully based on the referees' comments. The scientific quality of this paper is greatly improved. In light of the comprehensive revisions made, I recommend to accept this paper.

We appreciate all reviewers for their time spent to evaluate our manuscript and for their valuable comments. All the reviewers' concerns are addressed below in detail.

Reviewer #1 (Remarks to the Author):

In this version, the authors have provided evidence that addresses various concerns. From the given equations, it seems that the authors have developed a well-designed battery structure to achieve a CO₂ and H₂O assisted Na-O₂ battery system, in which the ORR can be considered to occur in an aqueous environment. So the ORR gets a higher potential compared with other organic electrolyte based Na-O₂ battery systems. Therefore, the authors should enhance the description of ORR in the manuscript to make it more reader-friendly before acceptance. Additionally, there are some points that need further revision.

1. In equations (1), (2), and (4), Na is shown as a reactant related to the formation of NaOH, as it only reacts at the anode. The authors should replace these equations with half-equations.

→ Thank you for your comments. We revised the equations with half-equations in the manuscript.

2. CV curves are still necessary, as they will not only display reduction reactions but also oxidation reactions. Both are important for rechargeable batteries.

→ We carried out cyclic voltammetry (CV) for the SE-based Na-air batteries (Fig. R1). The CV curve exhibited are consistent with the voltage profile in the constant current condition. The details of the CV measurement are explained in the revised manuscript and supporting information (Fig. S6).

Figure R1. Cyclic voltammogram of the SE-based Na-air battery at scan rate of 0.1 mV s⁻¹ under ambient air with 70% RH at 25 °C.

REVIEWERS' COMMENTS

Reviewer #1 (Remarks to the Author):

The authors have effectively addressed all the questions and made significant improvements to this paper. Therefore, it is now suitable for acceptance.